# R-MONet: Region-Based Unsupervised Scene Decomposition and Representation via Consistency of Object Representations

## Abstract

Decomposing a complex scene into multiple objects is a natural instinct of an intelligent vision system. Recently, the interest in unsupervised scene representation learning emerged and many previous works tackle this by decomposing scenes into object representations either in the form of segmentation masks or position and scale latent variables (i.e. bounding boxes). We observe that these two types of representation both contain object geometric information and should be consistent with each other. Inspired by this observation, we provide an unsupervised generative framework called R-MONet that can generate objects geometric representation in the form of bounding boxes and segmentation masks simultaneously. While bounding boxes can represent the region of interest (ROI) for generating foreground segmentation masks, the foreground segmentation masks can also be used to supervise bounding boxes learning with the Multi-Otsu Thresholding method. Through the experiments on CLEVR and Multi-dSprites datasets, we show that ensuring the consistency of two types of representation can help the model to decompose the scene and learn better object geometric representations.

## 1 Introduction

In recent years, supervised object detection and segmentation (He et al. (2017); Ren et al. (2015); Fan et al. (2019); Liao et al. (2001); Lin et al. (2017); Ronneberger et al. (2015)) have made great progress with the extensive human labels. However, these supervised methods are still unable to take the advantage of massive unlabeled vision data. Unsupervised learning of scene representation starts to become a key challenge in computer vision. The breakthrough (Burgess et al. (2019); Greff et al. (2019); Eslami et al. (2016); Crawford & Pineau (2019); Engelcke et al. (2020), Greff et al. (2017); Van Steenkiste et al. (2018); Pathak et al. (2016); Lin et al. (2020)) in the unsupervised scene decomposition and representation learning proves that a complex visual scene containing many objects can be properly decomposed without human labels. It proves that there is still much useful information that can be discovered in those unlabeled data.

Recent approaches to address the unsupervised scene decomposition and representation learning can be categorized into two groups: models which explicitly acquire disentangled position and scale (i.e. bounding boxes) representation of objects (Eslami et al. (2016); Crawford & Pineau (2019); Lin et al. (2020)) and models implicitly encode objects' geometric representation into segmentation masks or entangle it with object appearance representations (Burgess et al. (2019); Greff et al. (2019); Engelcke et al. (2020); Greff et al. (2017); Van Steenkiste et al. (2018)). In the former type of models, the scene is explicitly encoded into the object-oriented spatial encoding and appearance encoding. A decoder will generate the scene with explicitly defined object encoding for representation learning. This type of models cannot use rectangular bounding boxes to fully represent complex objects with flexible morphology. In the other type of models, the scene is decomposed into a finite number of object segmentation masks which can better represent complex objects with its pixel-to-pixel alignment. However, this type of models only use segmentation masks as the pixel-wise object mixture weights. They do not utilize the geometric information in the segmentation masks and still entangle object position and appearance representations in the scene generation step. Also, those models tend to decompose the entire scene in the image which does not use the locality benefit of objects.

Inspired by the observation that foreground segmentation masks and bounding boxes both contain object geometric information and should be consistent with each other, a method called R-MONet (Region-based Multiple Object Net) is proposed in this paper. R-MONet uses the spirit of MONet (Burgess et al. (2019)) and S4Net (Fan et al. (2019)) by using a single stage, non-iterative network (spatial attention module) for generating object geometric representations in both bounding boxes and segmentation masks. Then, a variational autoencoder (VAE) (Kingma & Welling (2013)) is used for encoding object appearance representations and regenerating the scene for training. To ensure the consistency between bounding boxes and foreground segmentation masks, the bounding boxes generated from spatial attention module is supervised with the pseudo bounding box generated by Multi-Otsu thresholding method (Liao et al. (2001)) on foreground segmentation masks. More than that, the foreground instance segmentation is only performed in the bounding box area instead of the full image to take advantage of the spatial locality and make scene generation less complex. The contributions of this paper are:

- We introduce an effective single stage, non-iterative framework to generate object geometric representations in both bounding boxes and segmentation masks for unsupervised scene decomposition and representation learning.

- We propose a self-supervised method that can better utilize object geometric information by ensuring the consistency between bounding boxes and foreground segmentation masks. This approach can improve the scene decomposition performance compared with the state-of-art.

- We design a new segmentation head that can preserve global context and prevent coordinate misalignment in small feature maps which improves the foreground segmentation performance.

## 2 RELATED WORKS

There many influential works (Burgess et al. (2019); Greff et al. (2019); Eslami et al. (2016); Crawford & Pineau (2019); Engelcke et al. (2020); Greff et al. (2017); Van Steenkiste et al. (2018); Pathak et al. (2016); Lin et al. (2020)) in unsupervised scene decomposition in recent years. Some models tend to explicitly factor an object representation into spatial and appearance encodings such as 'what', 'where', 'presence', etc. with the help of VAE (Kingma & Welling (2013)). Influential related models include AIR (Eslami et al. (2016)) and its successor SPAIR (Crawford & Pineau (2019)). AIR uses the Recurrent Neural Network as the encoder to decompose a complex scene into objects' representation but it suffers from the iteration speed. SPAIR improves its bounding box average precision and running speed by using Convolution Neural Network as the encoder to generate objects' representation in parallel. However, these models have not been tested on photorealistic 3D object dataset and bounding boxes can not fully represent flexible morphology like foreground segmentation masks.

The other type of models tend to decompose each object into its own representation without explicit positional encoding and use segmentation masks to mix object reconstruction. Influential models such as MONet (Burgess et al. (2019)) which leverages a UNet (Ronneberger et al. (2015)) variant as an iterative attention network for segmentation mask generation and Spatial Broadcast Decoder (Watters et al. (2019)) for representation learning via scene reconstruction. Spatial Broadcast Decoder replaces deconvolutional network with transform by tiling (broadcast) the latent vector across space, concatenate fixed X- and Y-"coordinate" channels. This decoder provides better disentanglement between positional and non-positional features in the latent distribution. IODINE (Greff et al. (2019)) tackles this problem with its amortized iterative refinement of foreground and background representation. However its iterative refinement process will heavily impact the speed of training and inference. GENESIS (Engelcke et al. (2020)) uses the similar idea as MONet but with different latent encoding in different iterative steps. These models all focus on decomposing the entire scene which does not leverage the spatial locality around each object.

SPACE (Lin et al. (2020)) is the closest to our work in spirit. This model leverages the encoder similar to SPAIR to process foreground objects in parallel with explicit positional encoding and adapts the segmentation masks for background modeling. However, different from R-MONet, it does not use the shared information in both bounding boxes and segmentation masks.

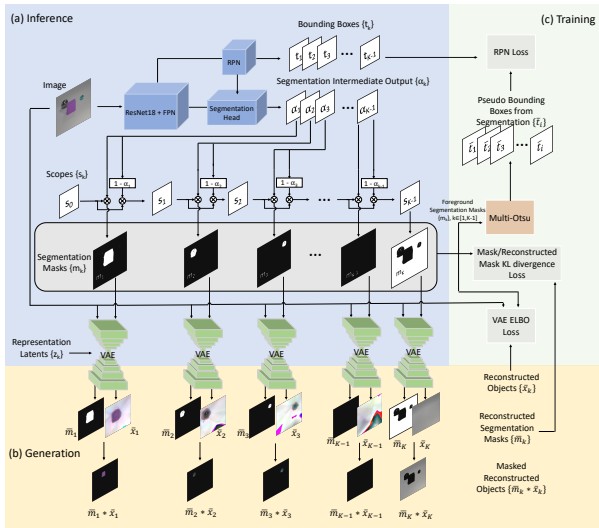

Figure 1: **End-to-End Architecture of R-MONet.** (a) A spatial attention network similar to S4Net receives the input image to generate foreground objects' bounding boxes $\{t_k\}$ and segmentation intermediate output $\{\alpha_k\}$. The scope $\{s_k\}$ represents the portion of the undecomposed scene. With the initial scope $s_0 = 1$, the stick-breaking process will transform $\{\alpha_k\}$ into the segmentation masks $\{m_k\}$ (K-1 masks for the foreground objects and 1 mask for the background) and keep $\sum_{k=1}^{K} m_k = 1$. VAE encoder takes the image and $\{m_k\}$ as the inputs to compute representation latents over segmentation and input image $\{z_k\}$. (b) VAE decoder reconstructs different objects $\{\bar{x}_k\}$ and their masks $\{\bar{m}_k\}$ with $\{z_k\}$ for representation learning. (c) To train the RPN in the inference network, $\{m_k\}_{k \in [1, K-1]}$ will be passed into the Multi-Otsu thresholding method to generate the pseudo bounding boxes $\{\bar{t}_i\}$. The image, $\{m_k\}$ and reconstructed objects $\{\bar{x}_k\}$ will form VAE ELBO loss. The KL divergence between $\{m_k\}$ and reconstructed segmentation masks $\{\bar{m}_k\}$ forces VAE to model the segmentation masks distribution in spatial attention module. The details of the loss is in Section 3.5.

It is worth mentioning that S4Net (Fan et al. (2019)) provides an end-to-end single shot object detection and instance segmentation framework simplified from Mask R-CNN (He et al. (2017)) for our spatial attention module. It uses the ROI Masking method inspired by the fact that the segmentation can benefit from the background features surrounding the foreground instance. In this method, an ROI mask is generated by enlarging the proposed region on the feature maps. The mask values inside the original proposed region are set to 1. The mask values outside the original proposed region but inside the enlarged proposed region are set to -1. The rest values on the mask are set to 0. The regional features are extracted by multiplying the ROI mask and the feature maps. This method is proved to have better segmentation compared with ROI Align (He et al. (2017)) in both their experiments and our use case (discussed in Section 4.1).

## 3 METHOD

In this section, the proposed model, R-MONet is described. The R-MONet will decompose the scene into K different components (K-1 for the foreground and 1 component for the background) and each of them is indexed by $k$. K also limits the max number of objects this model can detect. The network architecture of R-MONet is presented in Figure 1.

### 3.1 INFERENCE

**Inference of object geometric representations.** A spatial attention network parameterized by $\psi$ provides the object geometric representation in both segmentation masks $\{m_k\}$ and foreground object bounding box $\{t_k\}$ with the input image $x$. It is represented by equation 1. At the same

time, $\{\boldsymbol{m_k}\}$ is also the mixing probability in spatial Gaussian model. Since this mixing probability is learned by the network conditioned on $\boldsymbol{x}$, we refer to this distribution as $q_\psi(\boldsymbol{m_k}|\boldsymbol{x})$.

$$\{\boldsymbol{m_k}\}_{\boldsymbol{k}\in[\boldsymbol{1},\boldsymbol{K}]}, \{\boldsymbol{t_k}\}_{\boldsymbol{k}\in[\boldsymbol{1},\boldsymbol{K-1}]} = f_\psi(\boldsymbol{x}) \tag{1}$$

This spatial attention network is a convolutional neural network similar to S4Net (Fan et al. (2019)) with small adjustment. In this module, ResNet18 (He et al. (2016)) and Feature Pyramid Network (FPN) (Lin et al. (2017)) are used as the backbone. The feature maps are extracted for the following Region Proposal Network (RPN) (Ren et al. (2015)). The bounding boxes proposed by RPN will be filtered with Non-Maximum suppression (NMS) (Neubeck & Van Gool (2006)). Only ROIs with top K-1 prediction scores will be selected for further processing. The ROI Masking (Fan et al. (2019)) method then uses the ROI selected by RPN and transforms feature maps from the backbone to the segmentation head. The segmentation intermediate outputs $\{\boldsymbol{\alpha_k}\}$ (not segmentation masks) from the segmentation branch will be transformed into the segmentation masks $\{\boldsymbol{m_k}\}$ via the stick-breaking process (Sethuraman (1994)) to make sure $\sum_{k=1}^{K} \boldsymbol{m_k} = \boldsymbol{1}$ which represents all pixels are explained in the segmentation masks (softmax can also be used). The scope $\boldsymbol{s_k}$ represents the proportion of pixels still unexplained. The details about stick-breaking process is in Appendix A.

**Inference of object representation latents.** In the VAE encoder parameterised by $\phi$, we use variational inference to get an approximate latent posterior $q_\phi(\boldsymbol{z_k}|\boldsymbol{x}, \boldsymbol{m_k})$ which is a Gaussian distribution conditioned on both input image $\boldsymbol{x}$ and segmentation masks $\{\boldsymbol{m_k}\}$. As pointed out in MONet (Burgess et al. (2019)), conditioning on $\{\boldsymbol{m_k}\}$ can provide geometric heuristic information for probabilistic inference. Then, we sample from this posterior to infer object representation latents $\{\boldsymbol{z_k}\}$.

## 3.2 Improved Segmentation Head

During our experiments, we test two kinds of segmentation heads. Similar to S4Net, the first one extracts only conv3 layer feature maps for ROI Masking and adapts the similar segmentation head. We refer to it as R-MONet(Lite). R-MONet(Lite) can provide decent object geometric representation, but it can not generate sharp edges due to the missing of low level features. Inspired by the ROI Masking and the UNet (Ronneberger et al. (2015)), we propose an improved segmentation head which can extract feature maps from the region of interest (ROI) and keep the low level details with the skip connection. We name the proposed framework with the improved segmentation head as R-MONet(UNet) in the following sections. The architecture is shown in Appendix A. In this new head, feature maps from all layers except the conv4 and conv5 are applied ROI Masking. This design can extract the features in ROI but still preserve global context and prevent coordinate misalignment in small feature maps. Similar to the UNet, the top feature map from conv5 is passed into a small multilayer perceptron (MLP) (John Lu (2010)) and the feature maps from each layer have the skip connection to the same level in the segmentation branch. The quantitative comparison between R-MONet(Lite) and R-MONet(UNet) is shown in Section 4.

## 3.3 Generative Process

Assuming there are at most K objects (foreground and background) in the scene, we use K components in the input image $\boldsymbol{x}$ to represent K object representation latents $\{\boldsymbol{z_k}\}$. $\boldsymbol{z_k}$ is treated as independent Gaussian random variable with unit prior and the VAE decoder is shared across all components. The generative process is treated as a spatial Gaussian mixture model where each mixing component represents a single object. This means kth object representation $\boldsymbol{z_k}$ is passed into the VAE decoder parameterised by $\boldsymbol{\theta}$ to generate reconstructed kth object mask $\bar{m}_k$ (referred as $p_\theta(\bar{m}_k|\boldsymbol{z_k})$) and kth reconstructed object $\bar{\boldsymbol{x}}_k$ (i.e. pixel-wise mean). The kth corresponding mixing probability $p(component = k| \{\boldsymbol{m_k}\}) = \boldsymbol{m_k}$ (i.e. kth segmentation mask from inference) represents the probability pixels belong to component k. The input image likelihood given by the mixture model is:

$$p_\theta(\boldsymbol{x}| \{\boldsymbol{z_k}\}) = \sum_{k=1}^{K} \boldsymbol{m_k} p_\theta(\boldsymbol{x}|\boldsymbol{z_k})$$

$$= \boldsymbol{m_K}\mathcal{N}(\boldsymbol{x}; \bar{\boldsymbol{x}}_K, \boldsymbol{\sigma}_{bg}^2) + \sum_{k=1}^{K-1} \boldsymbol{m_k}\mathcal{N}(\boldsymbol{x}; \bar{\boldsymbol{x}}_k, \boldsymbol{\sigma}_{fg}^2) \tag{2}$$

where fixed variance $\boldsymbol{\sigma}^2_{fg}$ for K-1 foreground components and $\boldsymbol{\sigma}^2_{bg}$ for the background component. We use the spatial broadcast decoder (Crawford & Pineau (2019)) here for better disentanglement between positional and non-positional features in the latent distribution.

### 3.4 REGION BASED SELF-SUPERVISED TRAINING

Since foreground segmentation masks (The first K-1 segmentation masks) and bounding boxes (bboxs) both contain foreground object geometric information and should be consistent with each other, we can choose a simple unsupervised thresholding algorithm to separate the pixels of the foreground masks into several different classes based on the probability of being certain foreground object. With the separated classes in each component's foreground mask, we can generate pseudo positive bboxs for all anchors $\{\bar{\boldsymbol{t}_i}\}$ in the RPN where subscript $i$ is the anchor index. In our experiments, Multi-Otsu Thresholding (Liao et al. (2001)) algorithm can efficiently separate the foreground masks with a trivial overhead. More than that, models such as MONet, IODINE and GENESIS tends to decompose the entire scene which make the object representation hard to learn. Intuitively, only the surrounding area is necessary for the foreground object separation. Since feature maps outside the ROI Mask is set to 0, the object representation learning can focus only on the area selected by bboxs. This approach can provide a better segmentation from its surrounding area.

### 3.5 LOSS FUNCTION

The system is training end-to-end with the following loss:

$$L(\phi; \theta; \psi; \boldsymbol{x}) = -\log \sum_{k=1}^{K} \boldsymbol{m_k} p_\theta(\boldsymbol{x}|\boldsymbol{z_k}) + \beta \sum_{k=1}^{K} D_{KL}(q_\phi(\boldsymbol{z_k}|\boldsymbol{x}, \boldsymbol{m_k})||p(\boldsymbol{z_k}))$$
$$+ \gamma \sum_{k=1}^{K} D_{KL}(q_\psi(\boldsymbol{m_k}|\boldsymbol{x})||p_\theta(\bar{\boldsymbol{m}}_{\boldsymbol{k}}|\boldsymbol{z_k})) \quad (3)$$
$$+ \delta \sum_i p_i L_{cls}(\boldsymbol{p_i}, \bar{\boldsymbol{p}_i}) + \lambda \frac{1}{N_{reg}} \sum_i p_i L_{reg}(\boldsymbol{t_i}, \bar{\boldsymbol{t}_i})$$

The first term of the loss is the VAE decoder negative log-likelihood loss. This loss makes the reconstructed objects approximate to the scene inside the segmentation masks. The second term of the loss is the Kullback-Leibler (KL) divergence between the latent variable posterior distribution $q_\phi(\boldsymbol{z_k}|\boldsymbol{x}, \boldsymbol{m_k})$ and the latent variable prior distribution $p(\boldsymbol{z_k})$ (unit Gaussian prior). The first two terms are derived from the standard VAE's variational lower bound (ELBO) (Kingma & Welling (2013)).

The third term is the KL divergence between the distribution of network generating segmentation masks $q_\psi(\boldsymbol{m_k}|\boldsymbol{x})$ and the VAE's decoded segmentation masks distribution $p_\theta(\bar{\boldsymbol{m}}_{\boldsymbol{k}}|\boldsymbol{z_k})$. This term forces the VAE to generalize the distribution of the segmentation masks $\{\boldsymbol{m_k}\}$ coming from the spatial attention network.

The last two terms of loss are the bounding box (bbox) regression loss from the Regional Proposal Network(RPN) (Ren et al. (2015)). They penalize the difference between the pseudo bbox coordinates $\{\bar{\boldsymbol{t}_i}\}$ which from foreground segmentation masks and the bbox coordinates for all anchors $\{\boldsymbol{t_i}\}$ from the RPN. The penultimate term of the loss is a binary classification log loss between the predicted probability of anchor $\boldsymbol{i}$ being an object ($\boldsymbol{p_i}$) and the pseudo label of anchor $\boldsymbol{i}$ being positive or negative ($\bar{\boldsymbol{p}_i}$). The detail about assigning positive/negative labels to anchor $\boldsymbol{i}$ is described in Faster R-CNN (Ren et al. (2015)). The last term of the loss is a smooth $L_1$ loss (Girshick (2015)) between the RPN proposed bboxes and bboxes from foreground segmentation masks. The $N_{reg}$ in the last term is the number of anchors in an image. The last 4 terms of the loss are weighted by the tuneable hyperparameter $\beta$, $\gamma$, $\delta$ and $\lambda$.

## 4 RESULTS

We evaluate our model on two datasets:

**1) CLEVR** (Johnson et al. (2017)) contains photorealistic simple rendered 3D objects. The source images and labels come from the Multi-Object Datasets (Kabra et al. (2019)) and consist of 50000 images with a resolution of 240*320. We use the method proposed in MONet (Burgess et al. (2019)) to transform the images into 128*128. The dataset is divided into 48400 images for training and 1600 images for testing. As mentioned in (Greff et al. (2019)), only images with 3-6 objects (inclusive) is used. In the experiment on CLEVR, the number of components $K$ is set to 7 (6 slots for the foreground and 1 slot for the background) for all compared models (except SPACE).

**2) Multi-dSprites** (Matthey et al. (2017), Kabra et al. (2019)) consists of 61600 images with a resolution of 64*64. It contains 2D colourised sprites as the foreground objects and the grayscale background with uniform randomly brightness in each scene. Each scene contains 2-5 random sprites which vary in terms of shape (square, ellipse, or heart), color (uniform saturated colors), scale (continuous), position (continuous), and rotation (continuous). We divide this dataset into 60000 images for training and 1600 images for testing. In the experiment on Multi-dSprites, the number of components $K$ is set to 6 (5 slots for the foreground and 1 slot for the background) for compared models (except SPACE).

The compared models in the evaluation are given by:

- **MONet**, **IODINE**,: Our baselines in terms of ARI. These models do not generate foreground object position and scale representation (i.e. bbox).

- **MONet(ResNet18 + FPN)**: This is a variant of MONet. It replaces the UNet with the spatial attention network (ResNet18 + FPN + segmentation head) in R-MONet(Lite). Since it does not contain object detection branch and proposed self-supervised loss, it is used to prove that the performance gain of R-MONet(Lite) does not come from the difference of spatial attention networks.

- **MONet(ResNet18 + FPN + UNet)**: This is a variant of MONet. It replaces the UNet with the spatial attention network (ResNet18 + FPN + UNet like connection) in R-MONet(UNet). Since it does not contain object detection branch and proposed self-supervised loss, it is used to prove that the performance gain of R-MONet(UNet) does not come from the difference of spatial attention networks.

- **SPACE**: Our baselines evaluated with both ARI and mAP.

- **R-MONet(ROI Align)**: This is the R-MONet framework adapts the ROI Align and the segmentation head similar to Mask R-CNN.

- **R-MONet(Lite)**: This is the R-MONet framework adapts the ROI Masking and the segmentation head similar to Mask R-CNN.

- **R-MONet(UNet)**: This is the R-MONet framework adapts the ROI Masking and the improved segmentation head proposed in this paper.

**3) Quantitative Evaluation Metrics**: To quantify the decomposition and segmentation quality, we measure the similarity between the ground-truth foreground segmentation masks and the foreground segmentation masks output from the spatial attention network with the Adjusted Rand Index(ARI) (Rand (1971)). ARI measures the clustering similarity ranges from -1 to 1. The random labeling will have a ARI score close to 0. The ARI score will be close to 1 for perfect matching. The advantage of ARI is it can handle arbitrary permutations in the output and target clusters.

To quantify the quality of foreground object position and scale representation (i.e. bounding boxes), we use the mean average precision (mAP) (Lin et al. (2014)) from the MS-COCO metrics (Lin et al. (2014)). In this evaluation, all foreground objects will be treated as the same class.

### 4.1 RESULTS ANALYSIS AND ABLATION STUDY

**MONet vs R-MONet**

As shown in Table 1, both R-MONet(Lite) and R-MONet(UNet) surpass MONet on CLEVR dataset in terms of ARI. This is because MONet's loss does not explicitly restrict multiple objects with similar colors existing in the same mask. As we can see in Figure (2, 4, 8), MONet can not separate close objects with similar colors. More than that, since MONet performs segmentation on the entire image, it often suffers from small objects. When two objects are visually connected with each other,

Table 1: Segmentation performance comparison of MONet, MONet(ResNet18+FPN), IO-DINE, SPACE, MONet(ResNet18+FPN+UNet), R-MONet(ROI Align), R-MONet(Lite), R-MONet(UNet).

| Model | Dataset | ARI |
|---|---|---|
| MONet | CLEVR | 0.927 |
| MONet (ResNet18 + FPN) | CLEVR | 0.822 |
| MONet (ResNet18 + FPN + UNet) | CLEVR | 0.825 |
| IODINE | CLEVR | 0.962 |
| SPACE | CLEVR | 0.934 |
| R-MONet (ROI Align) | CLEVR | 0.878 |
| R-MONet (Lite) | CLEVR | 0.949 |
| R-MONet (UNet) | CLEVR | **0.981** |
| MONet | M-dSprites | 0.944 |
| MONet (ResNet18 + FPN) | M-dSprites | 0.607 |
| MONet (ResNet18 + FPN + UNet) | M-dSprites | 0.853 |
| IODINE | M-dSprites | 0.724 |
| SPACE | M-dSprites | 0.844 |
| R-MONet (ROI Align) | M-dSprites | 0.633 |
| R-MONet (Lite) | M-dSprites | 0.801 |
| R-MONet (UNet) | M-dSprites | **0.951** |

Table 2: The foreground object bounding boxes comparison (bounding box mean average precision) of SPACE, R-MONet(ROI Align), R-MONet(Lite), R-MONet(UNet).

| Model | Dataset | $mAP_{50:95}$ | $mAP_{50}$ | $mAP_{75}$ |
|---|---|---|---|---|
| SPACE | CLEVR | 0.592 | 0.863 | 0.679 |
| R-MONet (ROI Align) | CLEVR | 0.488 | 0.865 | 0.496 |
| R-MONet (Lite) | CLEVR | 0.611 | 0.935 | 0.730 |
| R-MONet (UNet) | CLEVR | **0.645** | **0.969** | **0.775** |
| SPACE | M-dSprites | 0.309 | 0.757 | 0.174 |
| R-MONet (ROI Align) | M-dSprites | 0.319 | 0.711 | 0.237 |
| R-MONet (Lite) | M-dSprites | 0.427 | 0.802 | **0.419** |
| R-MONet (UNet) | M-dSprites | **0.435** | **0.839** | 0.407 |

MONet may group them together as a single object even with different colors. This case is shown in Figure (5, 6). We can also find in Figure (9, 10), MONet may split a single object in multiple masks under certain lighting effects such as reflection or shadow. On the contrary, both R-MONet(Lite) and R-MONet(UNet) will not have this problem. Since segmentation of R-MONet is only performed inside ROI, the segmentation is more accurate. With the help of Multi-Otsu thresholding method and loss from pseudo bbox, proposed self-supervision will split the ROIs which contain multiple objects.

As shown in Table 1, R-MONet(UNet) still surpasses MONet while R-MONet(Lite)'s performance is worse than MONet on Multi-dSprites dataset. Comparing Multi-dSprites with CLEVR, the former dataset contains more sharp edges and more rectangular shapes. Since R-MONet(Lite) only extracts conv3 layer feature maps from FPN, the segmentation head can not generate sharp edges. With help of UNet like connection, MONet and proposed improved segmentation head in R-MONet(UNet) can produce segmentation masks with sharper edges and more complex shapes. Because Multi-dSprites is a 2D dataset, it does not have complex lighting effects, MONet is slightly better on this dataset. However, we can still see some problems in output of MONet such as multiple objects in the same mask (Figure 3, 12, 13) or single object split (Figure 11, 14, 15, 16).

Besides the segmentation performance, R-MONet converges faster than MONet during training due to the defects of MONet's loss. We use the ARI evaulated on test set as a measurement of convergence. Details can be found in Figure 18, Figure 19.

**MONet(ResNet18+FPN) and MONet(ResNet18+FPN+UNet)**

Through experiments on MONet(ResNet18+FPN) and MONet(ResNet18+FPN+UNet), we test whether the performance improvement is related to our framework or comes from the backbone difference. We change the input channel of the first convolution layer in ResNet18 from 3 to 4 and remove object detection branch. This change makes the backbone in our model be able to use as the recurrent attention module in MONet. As we can see in Table 1, after switching the UNet like backbone with (ResNet18+FPN) or (ResNet18+FPN+UNet), the performance downgrades in terms of ARI on both datasets. Because the UNet in original MONet is the same as MONet(ResNet18+UNet), the problem may come from the FPN. Since FPN adds feature maps instead of concatenation, it tends to mix features from different layers. This is not helpful for segmentation task. Even with different backbones, MONet(ResNet18+FPN) and MONet(ResNet18+FPN+UNet) still suffer from the same problems such as multiple objects in the same mask or single object split. In summary, combining object detection branch and proposed self-supervised method can effectively eliminate common problems of MONet.

**IODINE** As shown in Table 1, IODINE has higher quantitative segmentation performance than MONet on CLEVR dataset in terms of ARI. IODINE has much lower performance than MONet on Multi-dSprites dataset due to the soft edges in segmentation masks. This is expected and consistent with what is reported in its paper. From the qualitative results in Figure 2, IODINE seems to mix foreground objects with parts of background on CLEVR dataset. IODINE has better foreground separation on Multi-dSprites dataset but failed to generate accurate foreground masks. Our proposed model R-MONet(UNet) still surpasses IODINE on both datasets in terms of ARI.

**SPACE** As shown in Table 1 and Table 2, our proposed model R-MONet(Lite) and R-MONet(UNet) have better segmentation performance in terms of ARI and object localization performance in terms of mAP. This is consistent with the qualitative results. On CLEVR dataset, SPACE captures part of the object in Figure (7, 9). On Multi-dSprites dataset, SPACE tends to split the a single object into objects into multiple parts (Figure 3, 11, 12).

**ROI Masking vs ROI Align** Another ablation study we did is the comparison between R-MONet(ROI Align), R-MONet(Lite) and R-MONet(UNet). As seen in Table 1 and Table 2, the models using ROI Masking outperform the model using ROI Align in terms of ARI and mAP. This may be because the self-supervised bounding boxes generated by the Multi-Otsu method are not as accurate as the human labels in the supervised learning. ROI Masking is more robust than ROI Align when training with inaccurate pseudo bounding boxes because of the enlarged ROI. More than that, enlarged ROI can help the segmentation head find the object outside the proposed bounding boxes. When the bounding box only covers part of the foreground object, the enlarged foreground object mask can keep enlarging the proposed ROI until full object is covered with the pseudo bounding box label. Comparing R-MONet(Lite) and R-MONet(UNet), the UNet like segmentation head clearly outperforms the other segmentation head due to the skip connection and the added lower level feature maps.

**R-MONet(Lite) vs R-MONet(UNet)** No matter is quantitative result in terms of ARI and mAP or qualitative result, proposed improved segmentation head achieves better results. Clearly, proposed improved segmentation head can generate sharper edge and preserve more low level features than the segmentation head proposed in Mask-RCNN or S4Net. In terms of $mAP_{75}$ on Multi-dSprites dataset, R-MONet(Lite) outperforms R-MONet(UNet). This may be because of the highly occluded objects in Multi-dSprites dataset. A large occluded objects may be falsely detected as several small objects. In this case, a better segmentation does not guarantee a better object localization.

## 5 CONCLUSIONS

We propose R-MONet, a generative single stage framework for unsupervised scene decomposition and representation learning. By ensuring the consistency between foreground object bounding boxes and foreground segmentation masks, the model are able to decompose the foreground objects and learn better object geometric representations in the complex scene. After evaluation on CLEVR and Multi-dSprites, R-MONet achieves better quantitative decomposition performance than the state-of-art in terms of ARI and mAP. The future direction is to adapt this model with unsupervised visual representation learning on natural image datasets.

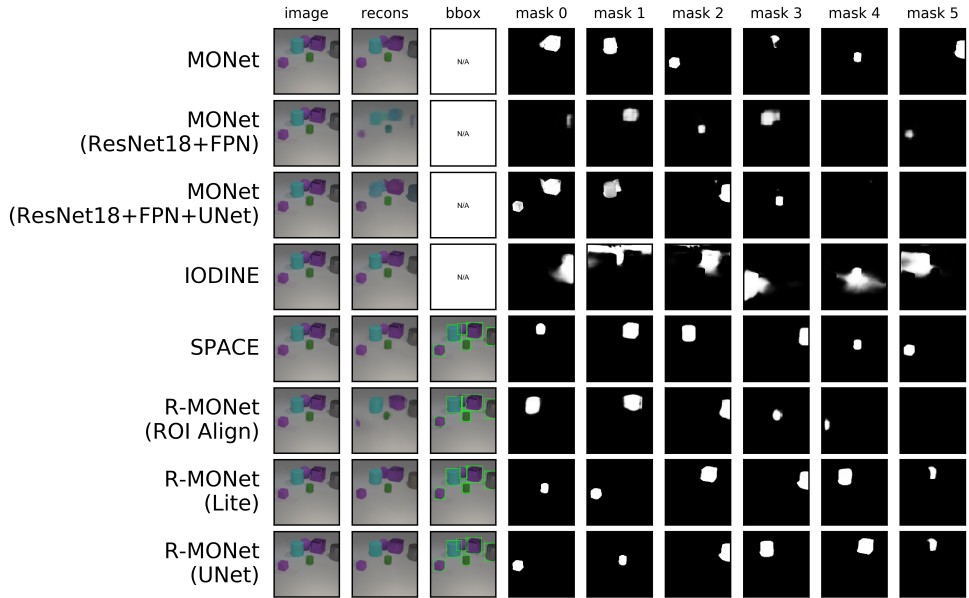

Figure 2: Qualitative comparison of scene reconstruction, bbox and foreground segmentation masks of MONet, MONet(ResNet18 + FPN), MONet(ResNet18 + FPN + UNet), IODINE, SPACE, R-MONet(ROI Align), R-MONet(Lite), R-MONet(UNet) on CLEVR dataset. More results can be seen in Appendix A

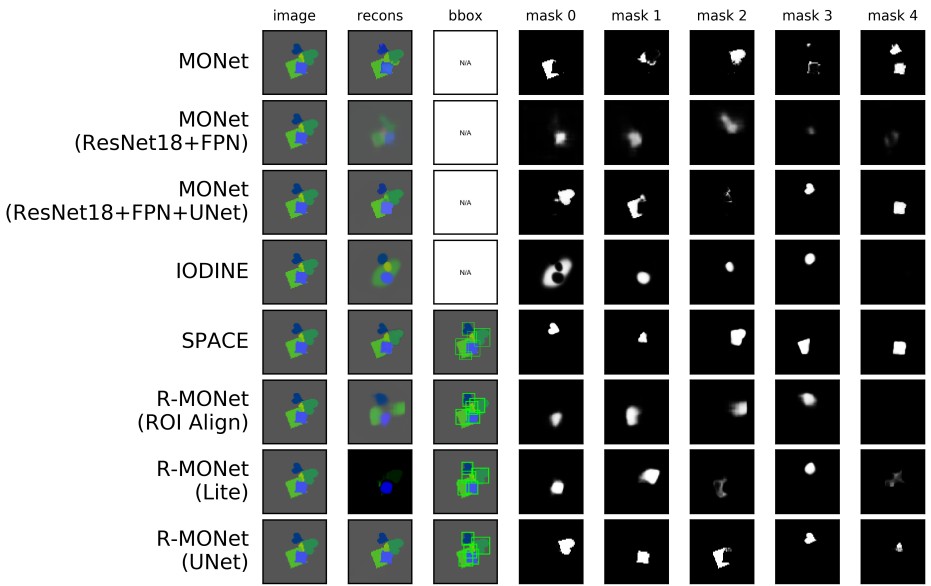

Figure 3: Qualitative comparison of scene reconstruction, bbox and foreground segmentation masks of MONet, MONet(ResNet18 + FPN), MONet(ResNet18 + FPN + UNet), IODINE, SPACE, R-MONet(ROI Align), R-MONet(Lite), R-MONet(UNet) on Multi-dSprites dataset. More results can be seen in Appendix A

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

# A   ADDITIONAL PLOTS

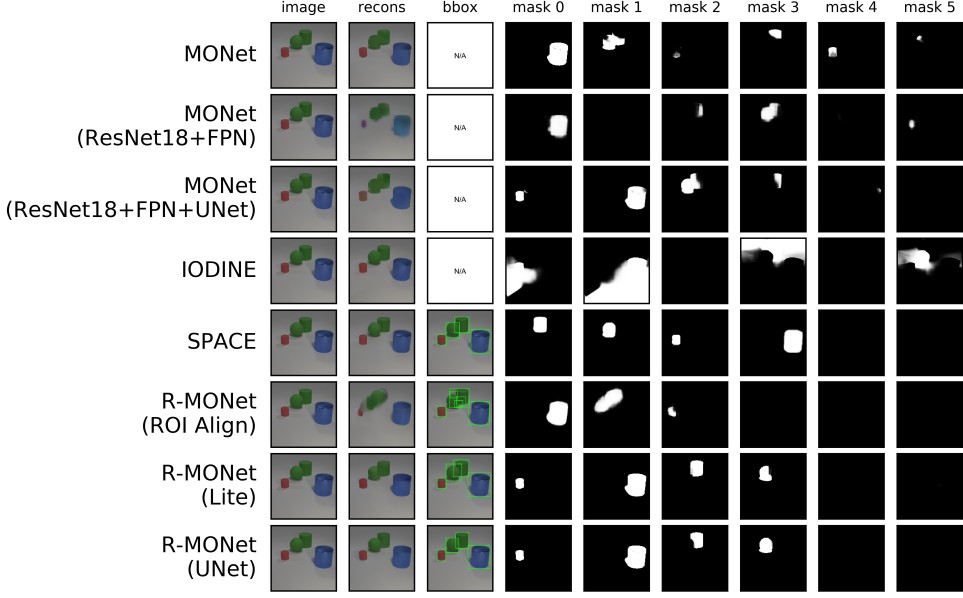

Figure 4: Additional qualitative comparison of scene reconstruction, bbox and foreground segmentation masks on CLEVR dataset.

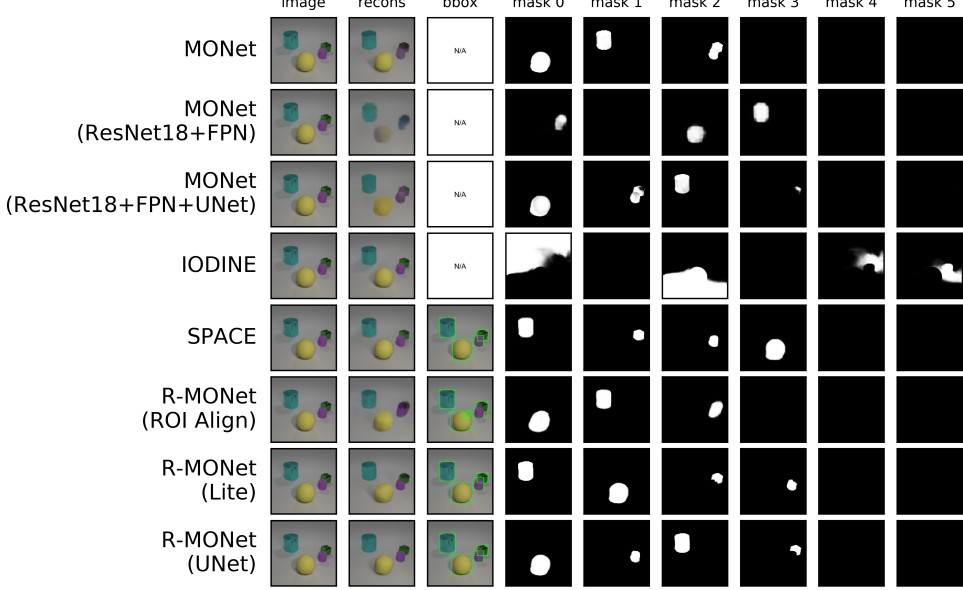

Figure 5: Additional qualitative comparison of scene reconstruction, bbox and foreground segmentation masks on CLEVR dataset.

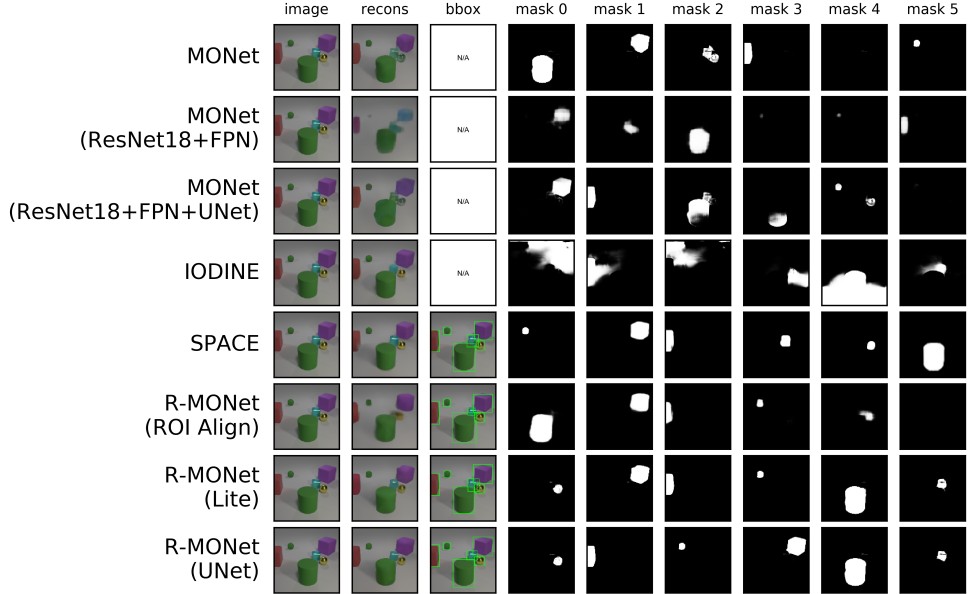

Figure 6: Additional qualitative comparison of scene reconstruction, bbox and foreground segmentation masks on CLEVR dataset.

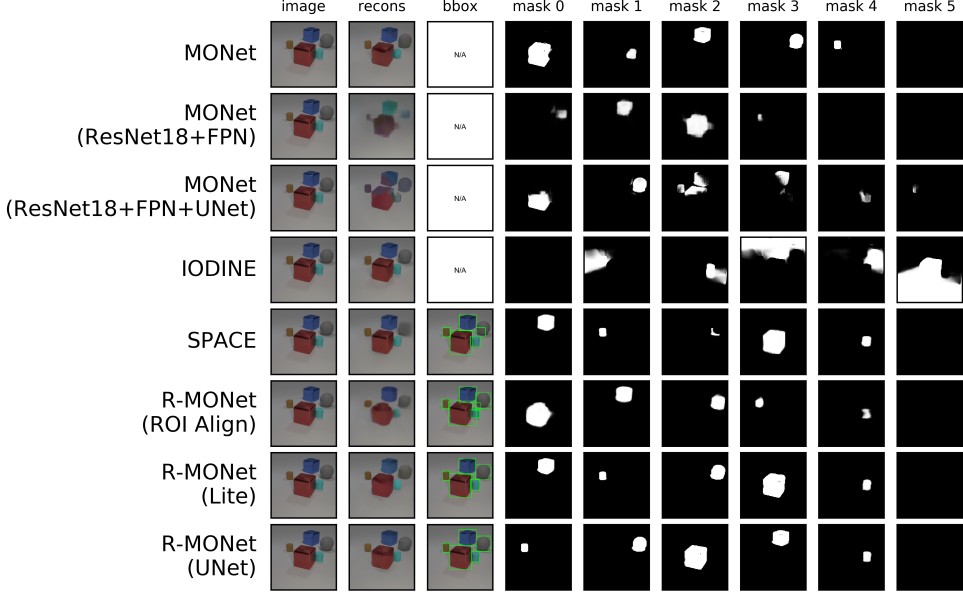

Figure 7: Additional qualitative comparison of scene reconstruction, bbox and foreground segmentation masks on CLEVR dataset.

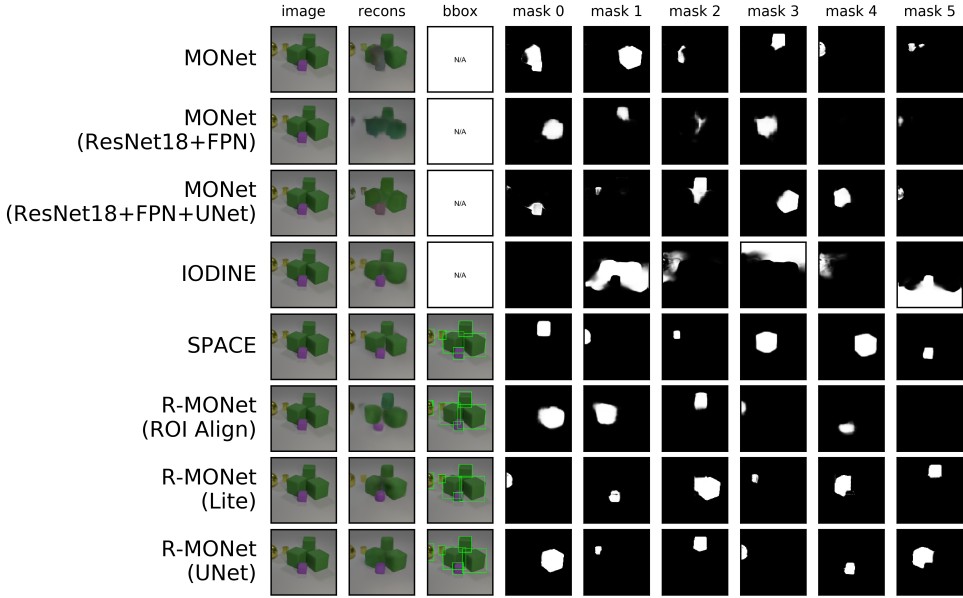

Figure 8: Additional qualitative comparison of scene reconstruction, bbox and foreground segmentation masks on CLEVR dataset.

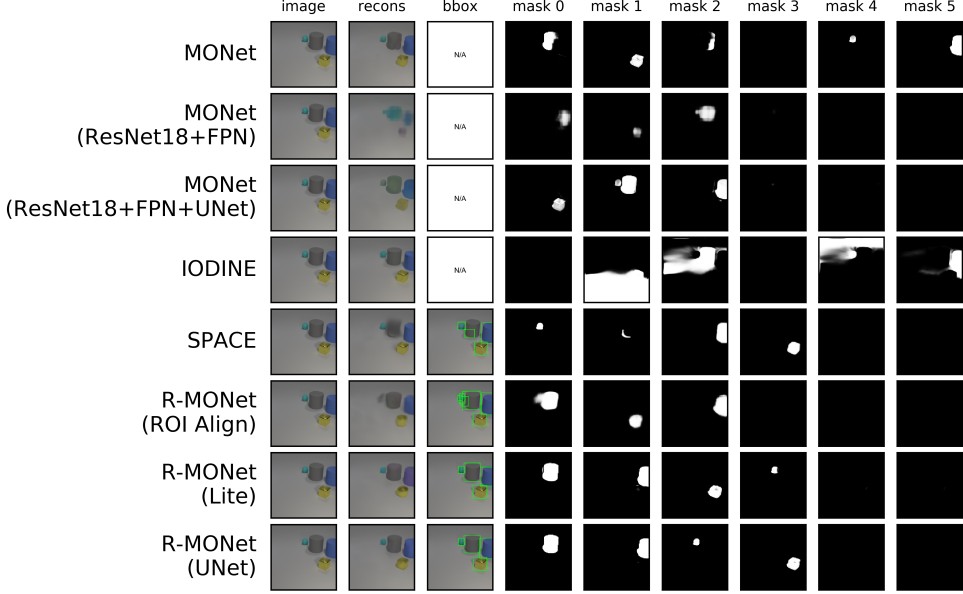

Figure 9: Additional qualitative comparison of scene reconstruction, bbox and foreground segmentation masks on CLEVR dataset.

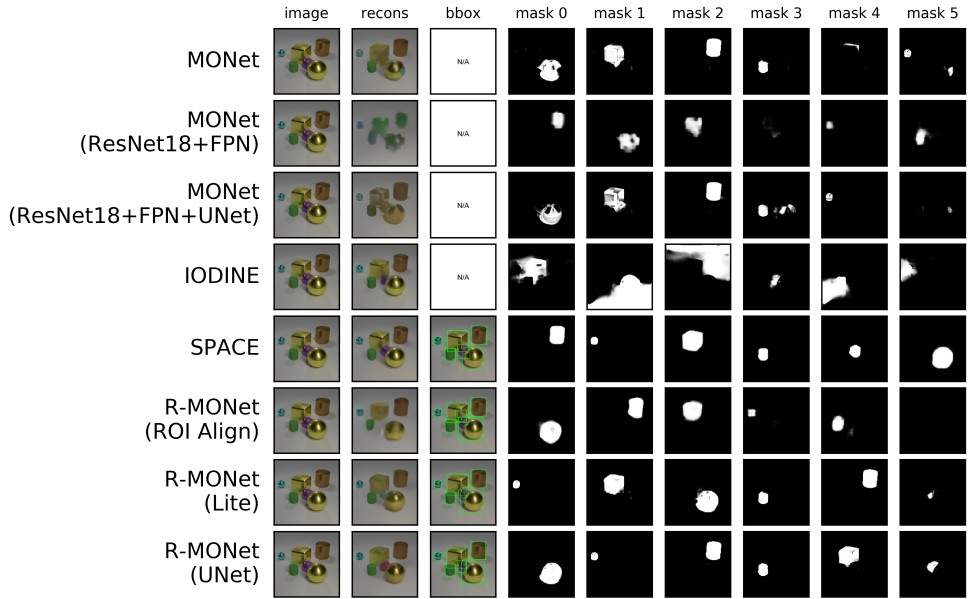

Figure 10: Additional qualitative comparison of scene reconstruction, bbox and foreground segmentation masks on CLEVR dataset.

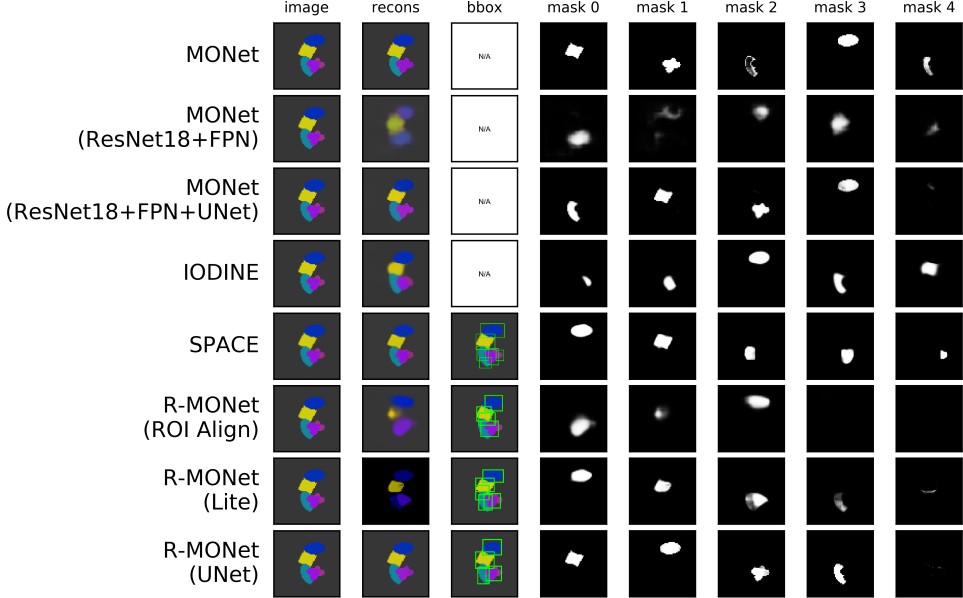

Figure 11: Additional qualitative comparison of scene reconstruction, bbox and foreground segmentation masks on Multi-dSprites dataset.

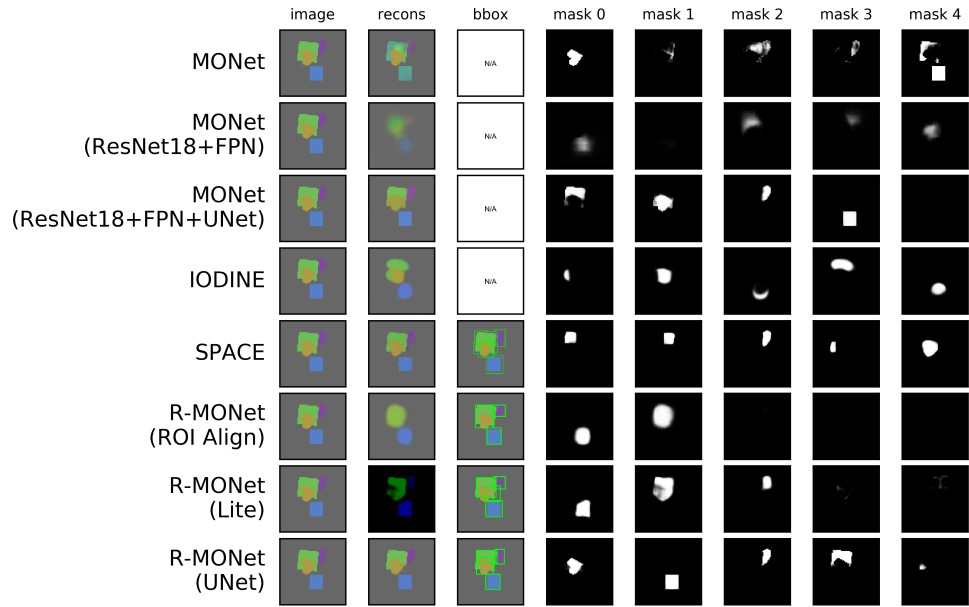

Figure 12: Additional qualitative comparison of scene reconstruction, bbox and foreground segmentation masks on Multi-dSprites dataset.

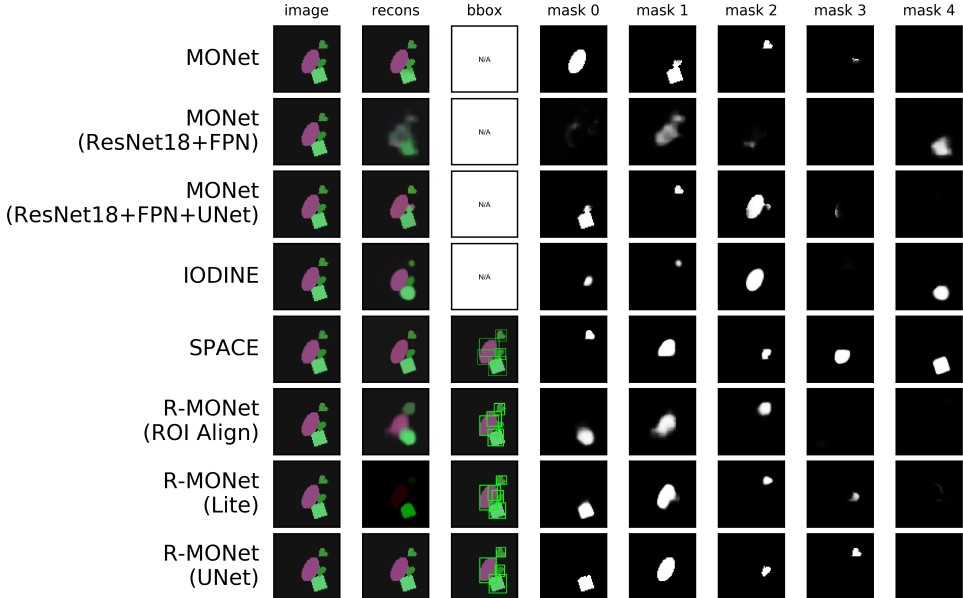

Figure 13: Additional qualitative comparison of scene reconstruction, bbox and foreground segmentation masks on Multi-dSprites dataset.

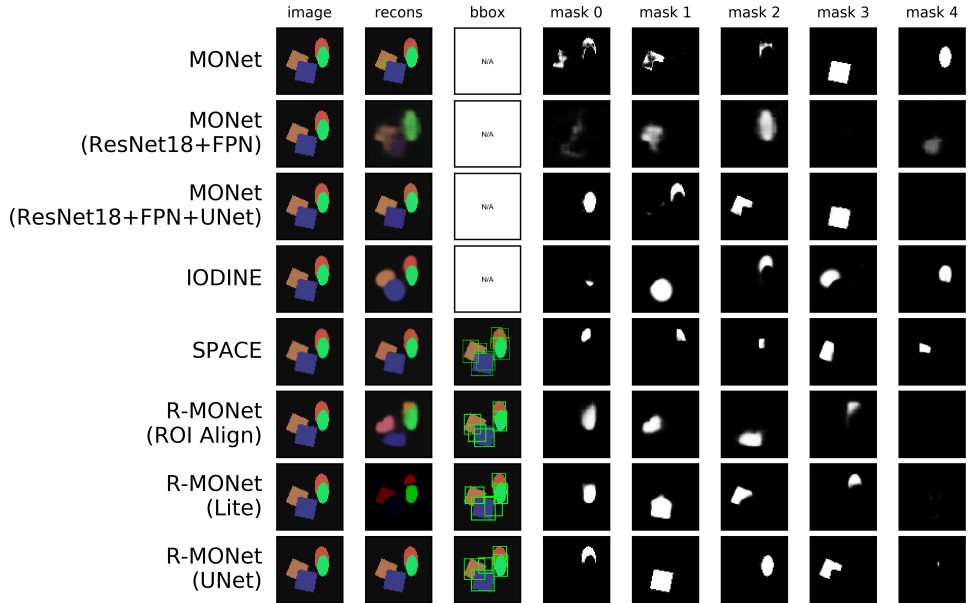

Figure 14: Additional qualitative comparison of scene reconstruction, bbox and foreground segmentation masks on Multi-dSprites dataset.

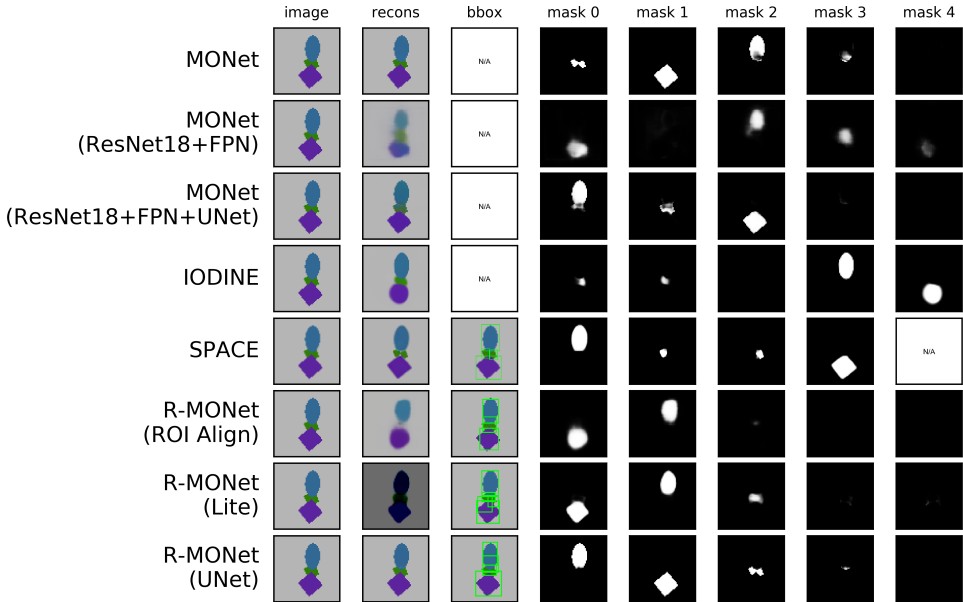

Figure 15: Additional qualitative comparison of scene reconstruction, bbox and foreground segmentation masks on Multi-dSprites dataset.

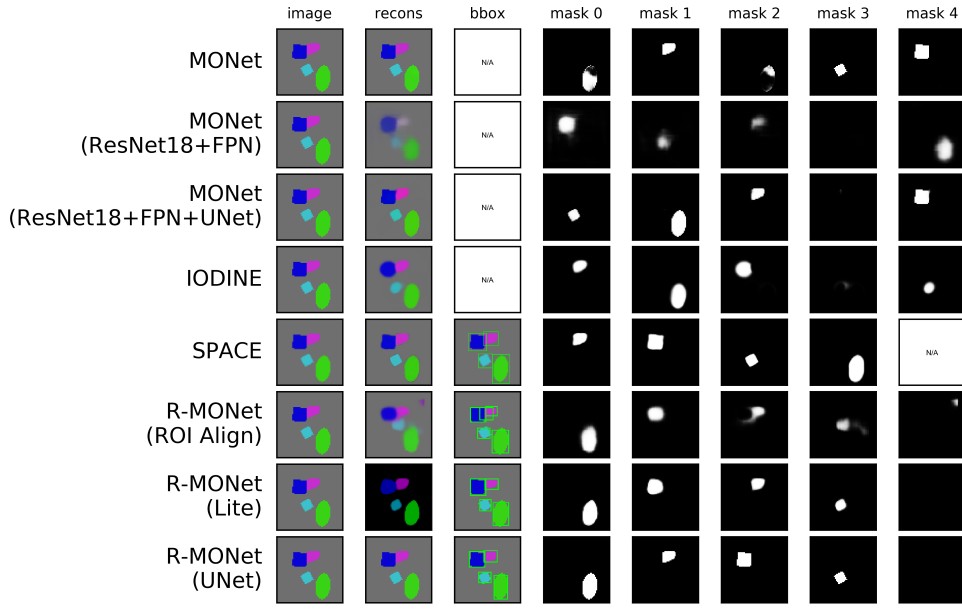

Figure 16: Additional qualitative comparison of scene reconstruction, bbox and foreground segmentation masks on Multi-dSprites dataset.

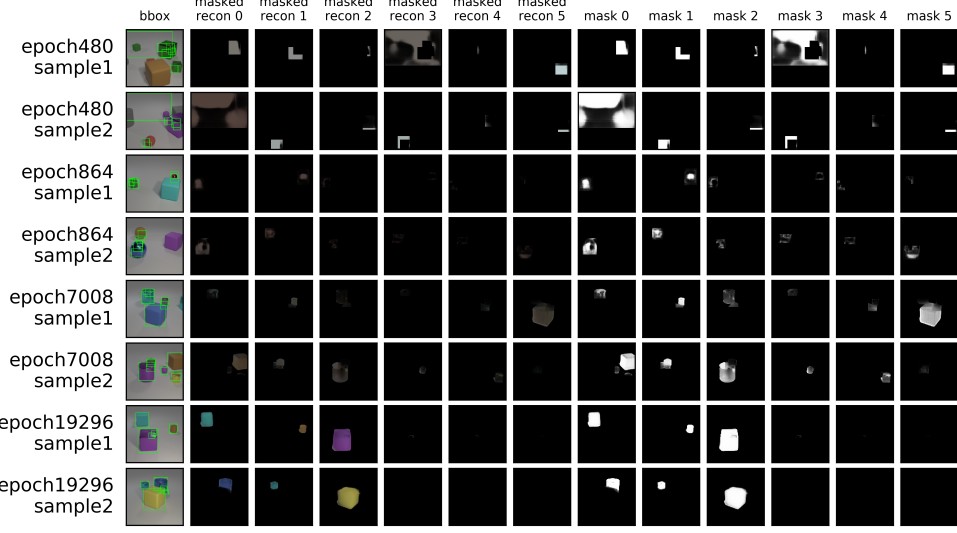

Figure 17: Qualitative comparison of R-MONet(UNet)'s masked object reconstruction, bbox and foreground segmentation masks at epoch 480, epoch 864, epoch 7008, epoch 19296 on CLEVR dataset. In the initial stage like epoch 480, ROIs are random across the image. The spatial attention network tends to learn segmentation inside the ROIs. After spatial attention network learns the rough segmentation masks (epoch 864), the pseudo ground truth bboxs generated from rough segmentation mask can guide object detection branch to find more accurate ROIs. In this stage, if segmentation masks contain more than one object, pseudo ground truth bbox will separate them with Multi-Otsu algorithm. At the middle stage (epoch 7008), the evolving segmentation masks help VAE to learn object appearance representations. In the last stage (epoch 19296), segmentation masks, bboxs and object appearance representations from VAE keep evolving at the same time.

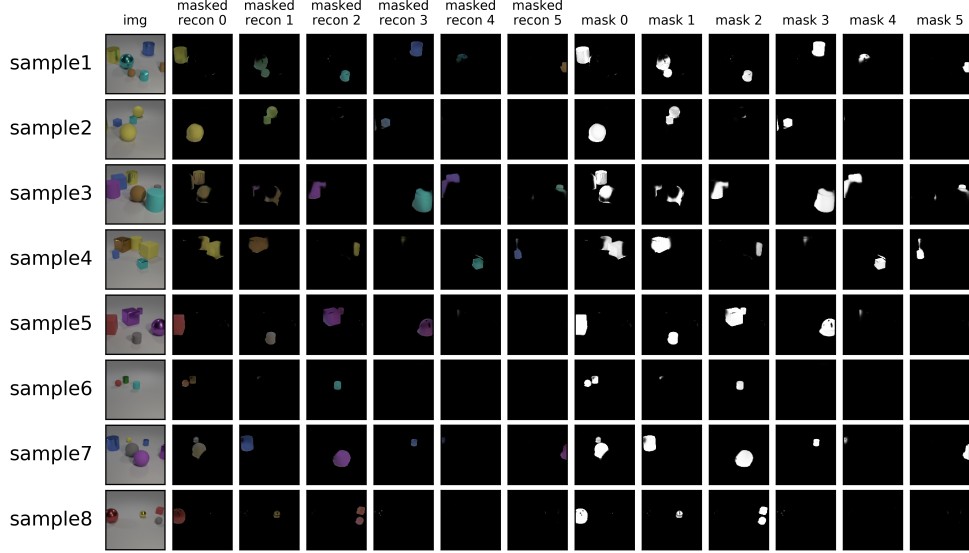

Figure 18: Qualitative comparison of MONet's masked object reconstruction, bbox and foreground segmentation masks at epoch 47040 on CLEVR dataset. During training, since the loss of MONet does not prevent multiple objects showing up in the same mask, MONet tends to distribute the objects with the same reconstructed color into the same mask. This problem may slow down MONet training.

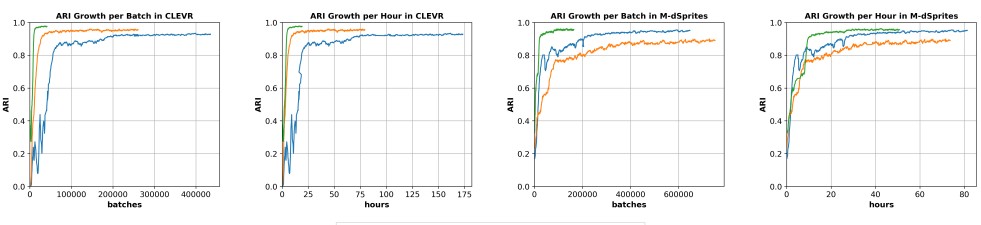

Figure 19: Quantitative training convergence speed comparison between MONet, R-MONet(Lite) and R-MONet(UNet). We compared the ARI performance growth speed in terms of batches and training time. On CLEVR dataset and Multi-dSprites dataset, R-MONet converges much faster than MONet. This may be because MONet uses the entire scene as the ROI and it is harder for attention network to obtain a proper segmentation for each object compared with regional ROI in R-MONet. Due to the better segmentation quality, R-MONet(UNet) converges even faster than R-MONet(Lite)

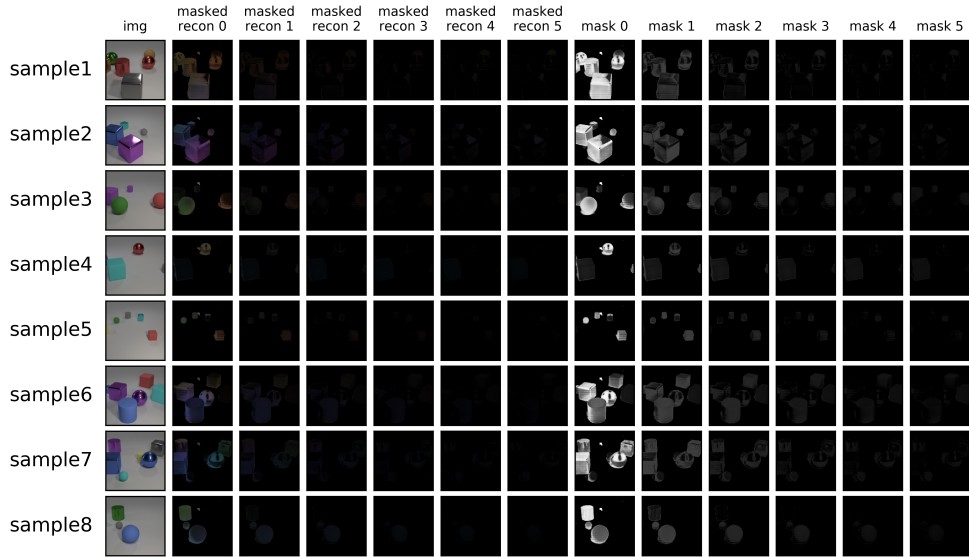

Figure 20: Qualitative comparison of R-MONet(UNet)'s (without the self-supervised loss and object detection branch) masked object reconstruction, bbox and foreground segmentation masks on CLEVR dataset. This model performs segmentation on the entire image and generates object masks in parallel. The ARI is nearly zero since all object segmentations are in one mask. The spatial attention module is good at segmenting objects from the background but unable to separate objects from each other. This proves that the loss of MONet does not prevent multiple objects from showing up in the same mask and the effectiveness of proposed self-supervised loss.

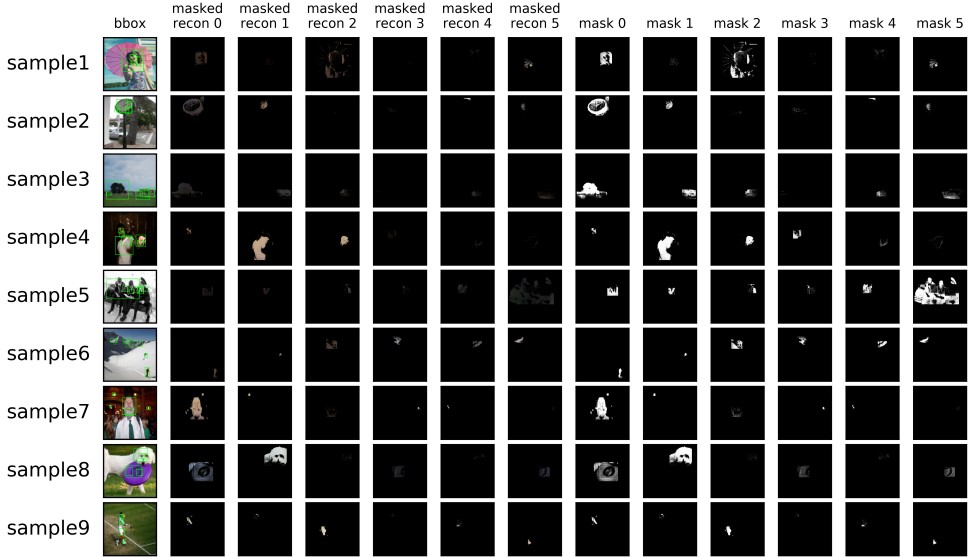

Figure 21: Qualitative comparison of R-MONet(UNet) masked object reconstruction, bbox and foreground segmentation masks on MS-COCO 2017 dataset (Lin et al. (2014)
). We use the pretrained ResNet18+FPN on ImageNet (Deng et al. (2009)) provided in torchvision (Marcel & Rodriguez (2010)). Unfortunately, the proposed model can not achieve the same performance as the supervised models. The VAE used can not generate complex object in the scene. It focuses more on the region which has high contrast with its surrounding areas.

# B IMPLEMENTATION DETAILS

**Stick-Breaking Process.** $\{\alpha_k\}$ is the segmentation intermediate outputs. $\{m_k\}$ is the segmentation masks (K-1 foreground and 1 background). $s_k$ represents the scope which is the proportion of pixels still unexplained.

$$s_0 = 1 \tag{4}$$

$$s_k = s_{k-1}(1 - \alpha_k), k \in [1, K-1] \tag{5}$$

$$m_k = s_{k-1}\alpha_k, k \in [1, K-1] \tag{6}$$

$$m_K = s_{K-1} \tag{7}$$

**VAE Decoder Negative Log Likelihood.**

$$
\begin{aligned}
-\log p_\theta(\boldsymbol{x}| \{\boldsymbol{z_k}\}) &= -\log \sum_{k=1}^{K} \boldsymbol{m_k} p_\theta(\boldsymbol{x}|\boldsymbol{z_k}) \\
&= -\log \boldsymbol{m_K} \mathcal{N}(\boldsymbol{x}; \bar{\boldsymbol{x}}_K, \boldsymbol{\sigma}_{bg}^2) - \log \sum_{k=1}^{K-1} \boldsymbol{m_k} \mathcal{N}(\boldsymbol{x}; \bar{\boldsymbol{x}}_k, \boldsymbol{\sigma}_{fg}^2) \\
&= -\log \boldsymbol{m_K} \frac{1}{\boldsymbol{\sigma}_{bg}\sqrt{2\pi}} \exp(-\frac{(\boldsymbol{x} - \bar{\boldsymbol{x}}_K)^2}{2\boldsymbol{\sigma}_{bg}^2}) \\
&\quad -\log \sum_{k=1}^{K-1} \boldsymbol{m_k} \frac{1}{\boldsymbol{\sigma}_{fg}\sqrt{2\pi}} \exp(-\frac{(\boldsymbol{x} - \bar{\boldsymbol{x}}_k)^2}{2\boldsymbol{\sigma}_{fg}^2})
\end{aligned}
\tag{8}
$$

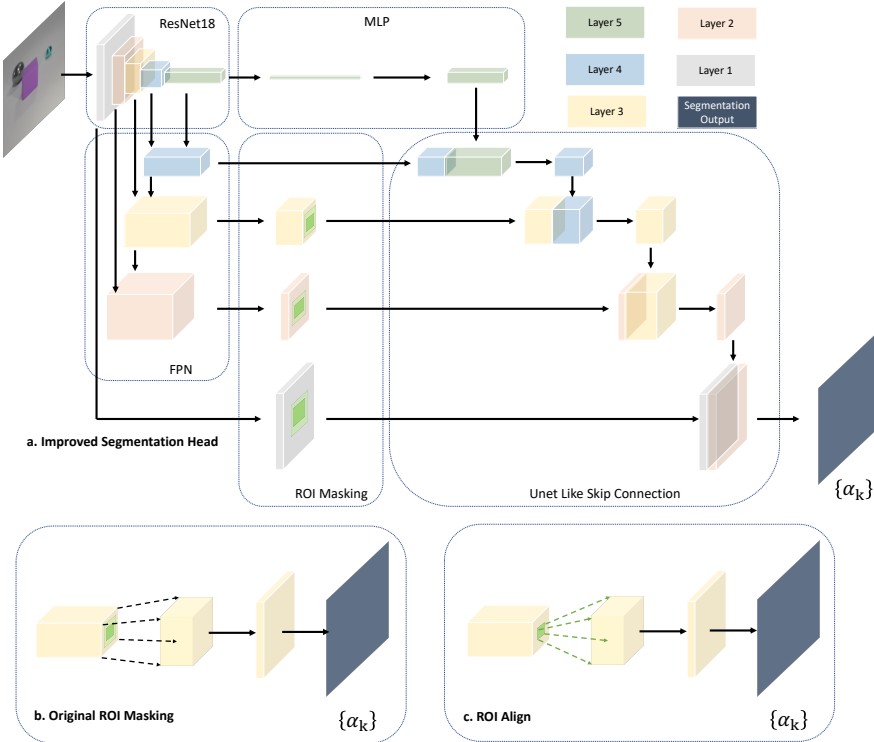

Figure 22: **The feature map flow of the improved segmentation head.** (a) The input image is passed into ResNet18 and conv{1-5} layers are extracted for the later use. The conv{2-5} layers are passed into FPN for the bounding boxes generation in RPN. After the FPN, the conv5 layer is passed into a MLP to compress its feature space. During the region feature selection, only conv{1-3} layers are transformed with ROI Masking method. The output of different layers are combined together like UNet to preserve low level features. (b) The original ROI Masking method enlarges the selected region by applying the ternary masks to the feature maps and it is only applied to the features from the conv3 layer. The transformed feature maps keep the conv3's dimensions. (c) The ROI Align method only extracts the feature maps in the selected area and transforms them into the fixed dimensions. The segmentation head is similar to the one used in the Mask R-CNN when adapting ROI Masking and ROI Align.

**ResNet18 + FPN + RPN.** The implementations of ResNet18, Feature Pyramid Network (FPN), Region Proposal Network (RPN) and bounding box loss used in RPN are adapted from torchvision (0.4.1) with some detail adjustment.

**MONet / Spatial Broadcast VAE.** The implementations of MONet and Spatial Broadcast VAE use some ideas from `github.com/baudm/MONet-pytorch`.

**Multi-Otsu Thresholding.** Multi-Otsu Thresholding method is a thresholding algorithm that is used to separate the pixels of an input image into several different classes by maximizing the between-class variance of pixel intensity (grey level). The implementations of Multi-Otsu Thresholding method and the method for generate bounding boxes with multiple thresholds are from scikit-image (0.16.2). Before applying Multi-Otsu method, the values under 0.001 is set to 0. The score for the pseudo ground truth bounding boxe $\bar{t}_i$ is the mean value of $m_k$ in the bounding box area.

**IODINE.** The implementation of IODINE is from `https://github.com/zhixuan-lin/IODINE`

**SPACE.** The implementation of SPACE is from `https://github.com/zhixuan-lin/SPACE`

### Hyperparameters on CLEVR and Multi-dSprites

For presented models, we performed a hyperparameter search and choose the results for the best settings. IODINE, SPACE, MONet, R-MONet(Lite) and R-MONet(UNet) are sensitive to VAE decoder scale (standard deviation). Other hyperparameters are robust in a reasonable range.

**R-MONet(Lite)**

| Module | Name | Value |
|---|---|---|
| CLEVR | input image dimension | (128, 128) |
| - | image normalization | - |
| - | batch size | 50 |
| - | optimizer | Adam |
| - | momentum | - |
| - | learning rate | 0.0001 |
| - | learning rate schedular | Cosine |
| - | learning rate decay interval | 200 epochs |
| - | learning rate reset decay | 0.05 |
| - | warmup | 10 epochs |
| ROI Masking | expand ratio $\sigma$ | 0.3 |
| RPN | anchor size | (16, 32, 64, 128, 256) |
| RPN | anchor aspect ratio | (0.5, 1.0, 2.0) |
| RPN | NMS threshold inside each layer | 0.3 |
| RPN | NMS threshold for output bbox | 0.3 |
| RPN | NMS top n | 6 |
| RPN | pre layer NMS top n | 100 |
| RPN | post layer NMS top n | 20 |
| RPN | foreground IOU threshold | 0.7 |
| RPN | background IOU threshold | 0.3 |
| RPN | batch size per image | 256 |
| RPN | positive fraction | 0.5 |
| RPN | prediction score threshold | 0.5 |
| Multi-Otsu | bbox minimum size | 5 |
| Multi-Otsu | bbox maximum h/w ratio | 3 |
| Multi-Otsu | bbox maximum w/h ratio | 3 |
| Multi-Otsu | NMS threshold | 0.7 |
| Multi-Otsu | number of bins | 5 |
| VAE | z dimension | 16 |
| VAE | background scale $\sigma_k$ | 0.06 |
| VAE | foreground scale $\sigma_k$ | 0.10 |
| VAE | component number K | 7 |
| - | $\beta$ | 0.5 |
| - | $\gamma$ | 0.5 |
| - | $\delta$ | 1 |
| - | $\lambda$ | 1 |

**R-MONet(UNet)**

| Module | Name | Value |
|---|---|---|
| CLEVR | input image dimension | (128, 128) |
| - | image normalization | - |
| - | batch size | 50 |
| - | optimizer | Adam |
| - | momentum | - |
| - | learning rate | 0.0001 |
| - | learning rate schedular | Cosine |
| - | learning rate decay interval | 50 epochs |
| - | learning rate reset decay | 0.1 |
| - | warmup | 5 epochs |
| ROI Masking | expand ratio $\sigma$ | 0.3 |
| RPN | anchor size | (16, 32, 64, 128, 256) |
| RPN | anchor aspect ratio | (0.5, 1.0, 2.0) |
| RPN | NMS threshold inside each layer | 0.3 |
| RPN | NMS threshold for output bbox | 0.3 |
| RPN | NMS top n | 6 |
| RPN | pre layer NMS top n | 100 |
| RPN | post layer NMS top n | 20 |
| RPN | foreground IOU threshold | 0.7 |
| RPN | background IOU threshold | 0.3 |
| RPN | batch size per image | 256 |
| RPN | positive fraction | 0.5 |
| RPN | prediction score threshold | 0.5 |
| Multi-Otsu | bbox minimum size | 5 |
| Multi-Otsu | bbox maximum h/w ratio | 3 |
| Multi-Otsu | bbox maximum w/h ratio | 3 |
| Multi-Otsu | NMS threshold | 0.7 |
| Multi-Otsu | number of bins | 5 |
| VAE | z dimension | 16 |
| VAE | background scale $\sigma_k$ | 0.06 |
| VAE | foreground scale $\sigma_k$ | 0.10 |
| VAE | component number K | 7 |
| - | $\beta$ | 0.5 |
| - | $\gamma$ | 0.5 |
| - | $\delta$ | 1 |
| - | $\lambda$ | 1 |

**MONet**

| Module | Name | Value |
|--------|------|-------|
| CLEVR | input image dimension | (128, 128) |
| - | image normalization | - |
| - | batch size | 50 |
| - | optimizer | Adam |
| - | momentum | - |
| - | learning rate | 0.0001 |
| - | learning rate schedular | Cosine |
| - | learning rate decay interval | 200 epochs |
| - | learning rate reset decay | 0.05 |
| - | warmup | 10 epochs |
| VAE | z dimension | 16 |
| VAE | background scale $\sigma_k$ | 0.06 |
| VAE | foreground scale $\sigma_k$ | 0.10 |
| VAE | component number K | 7 |
| - | $\beta$ | 0.5 |
| - | $\gamma$ | 0.5 |

**IODINE**

| Module | Name | Value |
|--------|------|-------|
| CLEVR | input image dimension | (128, 128) |
| - | image normalization | - |
| - | batch size | 20 |
| - | optimizer | Adam |
| - | learning rate | 0.0003 |
| - | component number K | 7 |
| - | iteration T | 5 |
| - | latent dimension | 64 |
| Refine Network | convolution output channel | 64 |
| Refine Network | convolution layers | 4 |
| Refine Network | mlp units | 256 |
| Refine Network | kernal size | 3 |
| Refine Network | stride | 2 |
| Decoder | convolution output channel | 64 |
| Decoder | decoder standard deviation $\sigma$ | 0.1 |
| Decoder | convolution layers | 4 |
| Decoder | kernal size | 3 |

**SPACE**

| Module | Name | Value |
|---|---|---|
| CLEVR | input image dimension | (128, 128) |
| - | image normalization | - |
| - | batch size | 48 |
| - | fg optimizer | RMSprop |
| - | bg optimizer | Adam |
| - | fg learning rate | 0.00001 |
| - | bg learning rate | 0.001 |
| - | grid | 8 |
| - | clip norm | 1 |
| fg VAE | fg standard deviation | 0.11 |
| fg VAE | z_pres start step | 4000 |
| fg VAE | z_pres end step | 10000 |
| fg VAE | z_pres start value | 0.1 |
| fg VAE | z_pres end value | 0.01 |
| fg VAE | z_scale mean start step | 0 |
| fg VAE | z_scale mean end step | 20000 |
| fg VAE | z_scale mean start value | -1.0 |
| fg VAE | z_scale mean end value | -2.0 |
| fg VAE | z_scale std value | 0.1 |
| fg VAE | tau start step | 0 |
| fg VAE | tau end step | 20000 |
| fg VAE | tau start value | 2.5 |
| fg VAE | tau end value | 0.5 |
| bg VAE | background component number K | 3 |
| bg VAE | bg standard deviation | 0.09 |
| - | boundary loss | true |
| - | boundary loss end step | 50000 |

**R-MONet(Lite)**

| Module | Name | Value |
|---|---|---|
| M-dSprites | input image dimension | (64, 64) |
| - | image normalization | (0.5, 0.5) |
| - | batch size | 64 |
| - | optimizer | Adam |
| - | momentum | - |
| - | learning rate | 0.0001 |
| - | learning rate schedular | Cosine |
| - | learning rate decay interval | 200 epochs |
| - | learning rate reset decay | 0.05 |
| - | warmup | 10 epochs |
| ROI Masking | expand ratio $\sigma$ | 0.3 |
| RPN | anchor size | (16, 32, 64, 128, 256) |
| RPN | anchor aspect ratio | (0.5, 1.0, 2.0) |
| RPN | NMS threshold inside each layer | 0.3 |
| RPN | NMS threshold for output bbox | 0.3 |
| RPN | NMS top n | 5 |
| RPN | pre layer NMS top n | 100 |
| RPN | post layer NMS top n | 20 |
| RPN | foreground IOU threshold | 0.7 |
| RPN | background IOU threshold | 0.3 |
| RPN | batch size per image | 256 |
| RPN | positive fraction | 0.5 |
| RPN | prediction score threshold | 0.5 |
| Multi-Otsu | bbox minimum size | 3 |
| Multi-Otsu | bbox maximum h/w ratio | 3 |
| Multi-Otsu | bbox maximum w/h ratio | 3 |
| Multi-Otsu | NMS threshold | 0.7 |
| Multi-Otsu | number of bins | 5 |
| VAE | z dimension | 16 |
| VAE | background scale $\sigma_k$ | 0.09 |
| VAE | foreground scale $\sigma_k$ | 0.11 |
| VAE | component number K | 6 |
| - | $\beta$ | 0.5 |
| - | $\gamma$ | 0.5 |
| - | $\delta$ | 1 |
| - | $\lambda$ | 1 |

**R-MONet(UNet)**

| Module | Name | Value |
|---|---|---|
| M-dSprites | input image dimension | (64, 64) |
| - | image normalization | - |
| - | batch size | 64 |
| - | optimizer | Adam |
| - | momentum | - |
| - | learning rate | 0.0001 |
| - | learning rate schedular | Cosine |
| - | learning rate decay interval | 50 epochs |
| - | learning rate reset decay | 0.1 |
| - | warmup | 5 epochs |
| ROI Masking | expand ratio $\sigma$ | 0.3 |
| RPN | anchor size | (16, 32, 64, 128, 256) |
| RPN | anchor aspect ratio | (0.5, 1.0, 2.0) |
| RPN | NMS threshold inside each layer | 0.3 |
| RPN | NMS threshold for output bbox | 0.3 |
| RPN | NMS top n | 5 |
| RPN | pre layer NMS top n | 100 |
| RPN | post layer NMS top n | 20 |
| RPN | foreground IOU threshold | 0.7 |
| RPN | background IOU threshold | 0.3 |
| RPN | batch size per image | 256 |
| RPN | positive fraction | 0.5 |
| RPN | prediction score threshold | 0.5 |
| Multi-Otsu | bbox minimum size | 3 |
| Multi-Otsu | bbox maximum h/w ratio | 3 |
| Multi-Otsu | bbox maximum w/h ratio | 3 |
| Multi-Otsu | NMS threshold | 0.7 |
| Multi-Otsu | number of bins | 5 |
| VAE | z dimension | 16 |
| VAE | background scale $\sigma_k$ | 0.09 |
| VAE | foreground scale $\sigma_k$ | 0.11 |
| VAE | component number K | 6 |
| - | $\beta$ | 0.5 |
| - | $\gamma$ | 0.5 |
| - | $\delta$ | 1 |
| - | $\lambda$ | 1 |

**MONet**

| Module | Name | Value |
| --- | --- | --- |
| M-dSprites | input image dimension | (64, 64) |
| - | image normalization | - |
| - | batch size | 64 |
| - | optimizer | Adam |
| - | momentum | - |
| - | learning rate | 0.0001 |
| - | learning rate schedular | Cosine |
| - | learning rate decay interval | 200 epochs |
| - | learning rate reset decay | 0.05 |
| - | warmup | 10 epochs |
| VAE | z dimension | 16 |
| VAE | background scale $\sigma_k$ | 0.09 |
| VAE | foreground scale $\sigma_k$ | 0.11 |
| VAE | component number K | 6 |
| - | $\beta$ | 0.5 |
| - | $\gamma$ | 0.5 |

**IODINE**

| Module | Name | Value |
| --- | --- | --- |
| M-dSprites | input image dimension | (64, 64) |
| - | image normalization | - |
| - | batch size | 64 |
| - | optimizer | Adam |
| - | learning rate | 0.0001 |
| - | component number K | 6 |
| - | iteration T | 5 |
| - | latent dimension | 16 |
| Refine Network | convolution output channel | 32 |
| Refine Network | convolution layers | 3 |
| Refine Network | mlp units | 128 |
| Refine Network | kernal size | 3 |
| Refine Network | stride | 2 |
| Decoder | convolution output channel | 32 |
| Decoder | decoder standard deviation $\sigma$ | 0.1 |
| Decoder | convolution layers | 5 |
| Decoder | kernal size | 3 |

**SPACE**

| Module | Name | Value |
|---|---|---|
| M-dSprites | input image dimension | (128, 128) (upscale from (64,64)) |
| - | image normalization | - |
| - | batch size | 48 |
| - | fg optimizer | RMSprop |
| - | bg optimizer | Adam |
| - | fg learning rate | 0.00001 |
| - | bg learning rate | 0.001 |
| - | grid | 8 |
| - | clip norm | 1 |
| fg VAE | fg standard deviation | 0.11 |
| fg VAE | z_pres start step | 4000 |
| fg VAE | z_pres end step | 10000 |
| fg VAE | z_pres start value | 0.1 |
| fg VAE | z_pres end value | 0.01 |
| fg VAE | z_scale mean start step | 0 |
| fg VAE | z_scale mean end step | 20000 |
| fg VAE | z_scale mean start value | -1.0 |
| fg VAE | z_scale mean end value | -2.0 |
| fg VAE | z_scale std value | 0.1 |
| fg VAE | tau start step | 0 |
| fg VAE | tau end step | 20000 |
| fg VAE | tau start value | 2.5 |
| fg VAE | tau end value | 0.5 |
| bg VAE | background component number K | 3 |
| bg VAE | bg standard deviation | 0.09 |
| - | boundary loss | true |
| - | boundary loss end step | - (keep boundary loss to the end of training) |
| - | boundary loss width | 2 |

