# OpenReview forum: "R-MONet: Region-Based Unsupervised Scene Decomposition and Representation via Consistency of Object Representations"
_ICLR.cc/2021/Conference — Reject_

### Official Review · AnonReviewer3 · 2020-10-26
**An extension of MONet for unsupervised scene decomposition with some interesting ideas but lack of sufficient explanations and experiments.**

**Rating:** 6
**Confidence:** 3

**Review:**

In this paper, the authors introduce a region-based approach for unsupervised scene decomposition. It extends the previous MONet by introducing the region-based self-supervised training. Instead of purely generating foreground masks in an RNN, they simultaneously predict the bounding boxes and segmentation masks using a Faster-RCNN-based framework. The supervision comes from the object reconstruction loss and the self-supervised loss of classification and regression of anchors in the RPN module. The experiments and comparisons are only conducted on the synthetic CLEVR and Multi-dSprites dataset.

[Paper strength]
- The paper is well motivated, and the proposed approach seems to be reasonable.
- The self-supervised idea is interesting that uses the segmentation mask to get the pseudo bounding box label for object detection, which could ensure the consistency of object mask and bounding box.

[Paper Weakness]
1. The self-supervision between segmentation masks and detection bounding boxes is the main contribution. While incorporating the self-supervision into MONet is meaningful and interesting, the overall novelty does not look significant.

2. Clarification of Methods:
- How to learn $m_k$ in a self-supervise way is unclear? MONet uses spatial attention to identify the most salient object one by one, which makes senses. But here you segment all the objects in one step. How could this be possible in an unsupervised way? From the example in Fig. 2, it looks R-MONet could pick out some small and far-away objects first, which is not intuitive.
- In Faster-RCNN, the positive/negative samples are selected by calculating the IoU threshold between the sampled bbox and the ground truth bbox. However, in this self-supervised approach, there is no ground truth bbox. Although the authors proposed to use the pseudo bbox from the segmentation mask $m_k$, how could this be reliable since $m_k$ is likely of poor quality, especially at the initial stage.
- The selected K value is unclear. In the original MONet, the spatial attention network is an RNN-like structure, they decompose the scene step-by-step. Therefore, they define the K steps. However, in this Faster-RCNN-based framework, the objects are selected in one step, how to select the K-1 objects in all proposals?

3. Results:
- Most of the results are with toy images. There is no result on real images.
- There is no result to really demonstrate the effectiveness of the self-supervised loss. The author should compare their R-MONet(UNet) with the baseline of R-MONet(UNet) w/o the self-supervised loss, i.e. removing the object detection branch. Another missing baseline is MONet(UNet).
- In Table 1, the MONet (ResNet18+FPN) is 10 percent lower than the original MONet. Does this means the network structure has a greater influence on the performance than the self-supervision component.
- In Table 1, the R-MONet(Lite) performs worse. Once again, I guess this poor performance comes from the network structure, as the input image is 64*64, the Resnet downsamples the image to a very low resolution which losses the spatial information.
- The visual results cannot demonstrate the advantage of the proposed approach. For example, in Figure 3, the visual performance of MONet and R-MONet(UNet) are quite similar.

---
---
Update:
In general, I am happy with the authors' responses. They did show the advantage of the introduced self-supervised loss. Although the self-supervised loss is intuitive, incorporating it into MONet is non-trivial and it does outperform MONet. Despite that such self-supervised works are hard to work on real scenes, this paper does have some merits. I am willing to increase my rating.

---

> ### Author Response · Authors · 2020-11-18
> **Response to AnonReviewer3 (1/3)**
>
> 1. "How to learn $m_k$ in a self-supervise way is unclear? MONet uses spatial attention to identify the most salient object one by one, which makes sense. But here you segment all the objects in one step. How could this be possible in an unsupervised way? From the example in Fig. 2, it looks R-MONet could pick out some small and far-away objects first, which is not intuitive."
>
> R-MONet generates ROIs and segmentation masks in parallel since we do not assume the objects in the scene are dependent. The recurrent attention structure in MONet is not necessary. Saliency objects can be captured in the local region of an image without having information about the rest of the scene. The ability to detect objects in a given ROI can be transferred well to detecting objects in other sub-regions. After the inference network generates segmentation masks, it passed them into the VAE to learn object appearance representation.
>
> 2. "In Faster-RCNN, the positive/negative samples are selected by calculating the IoU threshold between the sampled bbox and the ground truth bbox. However, in this self-supervised approach, there is no ground truth bbox. Although the authors proposed to use the pseudo bbox from the segmentation mask $m_k$, how could this be reliable since $m_k$ is likely of poor quality, especially at the initial stage."
>
> We add Fig.17 to visualize the training of R-MONet during the initial stage. At the very early stage like epoch 480, ROIs are random across the image. The spatial attention network tends to learn segmentation inside the ROIs. After the spatial attention network learns the rough segmentation masks, the pseudo ground truth bboxs generated from rough segmentation mask can guide object detection branch to find more accurate ROIs.  In this stage, if segmentation masks contain more than one object, pseudo ground truth bbox will separate them with Multi-Otsu algorithm. At the middle stage (epoch 7008), the evolving segmentation masks help VAE to learn object appearance representations. In the last stage (epoch 19296), segmentation masks, bboxs and object appearance representations from VAE keep evolving at the same time.
>
> 3. "The selected K value is unclear. In the original MONet, the spatial attention network is an RNN-like structure, they decompose the scene step-by-step. Therefore, they define the K steps. However, in this Faster-RCNN-based framework, the objects are selected in one step, how to select the K-1 objects in all proposals?"
>
> After RPN, only ROIs with top K-1 prediction scores will be selected for further processing. The K-1 in R-MONet represents the max number of objects it will detect. We set the same K as MONet since we want to compare two models fairly. A good model should be able to capture objects with the least amount of ROIs. We choose 0.5 for the prediction score threshold to filter out void ROIs. We can see in the qualitative comparison figures, the bounding boxes are filtered with the threshold(sometimes less than K-1). We still keep the blank masks in the figures to make sure visualization is consistent.
>
> 4. "Most of the results are with toy images. There is no result on real images."
>
> The datasets we used is widely adopted by many related works such as MONet and IODINE. Comparing with previous models like AIR, SPAIR, GENESIS, and SPACE, the CLEVR dataset we use contains complex lighting effects and relatively closer to real images. However, we want to mention that the unsupervised scene decomposing task on the real image dataset is still a challenging problem and none of the previous models as we know can achieve acceptable performance on these datasets. In `Sec.5` of the related work `IODINE[8]`(accepted in ICML2019): "IODINE groups ImageNet not into meaningful objects but mostly into regions of similar color". In `SPACE[19]`(accepted in ICLR2020): "Interesting future directions are to ... and to improve the model for natural images". Experiment only on synthetic datasets is widely adopted by SOTA(State-of-the-Art) works.
>
> 5. "There is no result to really demonstrate the effectiveness of the self-supervised loss. The author should compare their R-MONet(UNet) with the baseline of R-MONet(UNet) w/o the self-supervised loss, i.e. removing the object detection branch. Another missing baseline is MONet(UNet)."
>
> We add experiment MONet(ResNet18+FPN+UNet) (equals to R-MONet(UNet) w/o the self-supervised loss and object detection branch) in the revised paper. Original MONet is basically MONet(ResNet18+UNet). We believe the MONet(UNet) you mentioned is the same as MONet(ResNet18+FPN+UNet). MONet(ResNet18+FPN+UNet)  performs significantly worse than MONet. FPN has a negative effect on segmentation by adding feature maps from different levels.

---

> > ### Comment · AnonReviewer3 · 2020-11-22
> > **Missing baselines**
> >
> > Thanks the authors for the detailed reply. Here are some additional comments.
> >
> > 1. MONet(ResNet18+FPN+UNet) is NOT equal to R-MONet(UNet) w/o the self-supervised loss and object detection branch, as you mentioned that  MONet is based on iterative spatial attention while R-MONet is based on non-iterative attention. You need to show:  * What's the performance for R-MONet(UNet) w/o the self-supervised loss and object detection branch?
> > * If possible, what's the performance of MONet + the self-supervised loss and object detection branch?
> >
> > 2 Although others did not have the results in real images, it would be much convincing if you could show it. At least we know when the method works and when it fails.

---

> > > ### Author Response · Authors · 2020-11-24
> > > **Response to "Missing baselines"**
> > >
> > > 1. We added the qualitative result of R-MONet(UNet) (without the self-supervised loss and object detection branch) on CLEVR dataset in the latest revision Fig.20. This model performs segmentation on the entire image and generates object masks in parallel. The ARI is 0(less than 0.0001) since all object segmentations are in one mask. The spatial attention module is good at segmenting objects from the background but unable to separate objects from each other. This proves that the loss of MONet does not prevent multiple objects from showing up in the same mask and the effectiveness of proposed self-supervised loss.
> > >
> > > 2. Unfortunately, we are unable to finish the experiment of MONet + the self-supervised loss and object detection branch before the deadline. The running training will take at least 3 days.
> > >
> > > 3. We add the qualitative result of R-MONet(UNet) results on MS-COCO 2017 dataset in the latest revision Fig.21. We use the pretrained ResNet18+FPN on ImageNet provided in torchvision. Unfortunately, the proposed model can not achieve the same performance as the supervised models. The VAE used can not generate complex objects in the scene. It focuses more on the region which has high contrast with its surrounding areas.

---

> ### Author Response · Authors · 2020-11-18
> **Response to AnonReviewer3 (2/3)**
>
> 6. "In Table 1, the MONet (ResNet18+FPN) is 10 percent lower than the original MONet. Does this means the network structure has a greater influence on the performance than the self-supervision component."
>
> MONet (ResNet18+FPN) does not have object detection branch and self-supervised loss. It replaces UNet in the original MONet with ResNet18+FPN+(Segmentation head similar to Mask-RCNN). Since the results of MONet (ResNet18+FPN) are worse than the original MONet, we prove that the backbone difference can not explain the performance gain of R-MONet.
>
> 7. "In Table 1, the R-MONet(Lite) performs worse. Once again, I guess this poor performance comes from the network structure, as the input image is 64*64, the Resnet downsamples the image to a very low resolution which losses the spatial information."
>
> The R-MONet(Lite) performs the same even with upscaled images(128*128) on the Multi-dSprites dataset(did not report in the paper because it seems to be trivial). Since the objects in Multi-dSprites contains sharp edges and complex shapes,  feature maps from the conv3 layer lose too many low-level details. We have a similar conclusion as what is mentioned in Sec4.1 in IODINE. If we look at the boundary of segmentation on the CLEVR dataset, R-MONet(UNet) (with improved segmentation head)'s segementation is sharper than R-MONet(Lite). Due to the rectangular segmentation masks, R-MONet(Lite) can achieve decent performance on CLEVR.
>
> 8. "The visual results cannot demonstrate the advantage of the proposed approach. For example, in Figure 3, the visual performance of MONet and R-MONet(UNet) are quite similar."
>
> Thanks for pointing out. We add more qualitative comparison figures in the revised paper and add some analysis in section 4.1. MONet often split a single object into different masks or put multiple objects into the same mask since MONet's loss does not explicitly restrict multiple objects with similar colors existing in the same mask. Besides that, since MONet performs segmentation on the entire image each time, it will increase the complexity of segmentation and hurt the segmentation quality, especially for connected small objects.

---

> ### Author Response · Authors · 2020-11-18
> **Response to AnonReviewer3 (3/3)**
>
> 9. "The self-supervision between segmentation masks and detection bounding boxes is the main contribution. While incorporating the self-supervision into MONet is meaningful and interesting, the overall novelty does not look significant."
>
> **The contribution of R-MONet and difference from MONet**
> 1. We want to highlight that R-MONet is entirely different from MONet in terms of both the inference part and the training part. The only common structure is the generation part.
> 2. Regarding the inference part, MONet is designed to use a recurrent attention module to extract object' masks one by one. Also, MONet performs segmentation on the entire image each time. It will increase the complexity of segmentation and hurt the segmentation quality, especially for connected small objects. We discuss this problem in the revised result analysis section. On the contrary, R-MONet extracts object' masks in parallel since R-MONet does not assume the objects in the scene are dependent. Regarding segmentation, R-MONet only performs segmentation inside ROI by introducing object detection branch. Since the segmentation is only performed around ROI, the segmentation quality is better and be able to eliminate the problems of MONet. Besides that, R-MONet can perform both unsupervised object detection and unsupervised instance segmentation (in a way entirely different from SPACE) while MONet can only perform unsupervised instance segmentation. We do not agree R-MONet is a slight variation of MONet just like Mask-RCNN is not a slight variation of UNet.
> 3. Regarding the training part, we add another supervision from localization inconsistency between predicted bbox and instance masks. We find MONet often split a single object into different masks or put multiple objects into the same mask (discussed in the revised result analysis section). This is caused by the defect of MONet's loss. MONet's loss does not explicitly restrict multiple objects with similar colors existing in the same mask. On the contrary, R-MONet will not have this problem. With the help of the Multi-Otsu thresholding method and loss from pseudo bbox, proposed self-supervision will split the ROIs which contain multiple objects. It is worth mentioning that previous jointly learning detection and instance segmentation(Mask-RCNN) only share some spirits with us. It is essentially different tasks receiving their own ground truth. Differently, our approach makes detection and instance segmentation supervise each other.
> 4. People may argue that the inference structure(ResNet18, FPN, RPN) we used is not new. At first, we want to point out that the network structure directly from the supervised learning topic does not perform very well in our tasks. R-MONet(Lite) (with the same inference structure as S4Net) does not surpass all our baselines. The old network structure needs to be carefully adapted and evaluated through experiments (well-known ROI Align significantly hurts the result). The best performance can only be achieved with our improved segmentation head (R-MONet(UNet)). This contribution is not trivial. Besides that, We want to highlight that the inference structure does not work without our proposed self-supervised method since object detection branch can not be trained without pseudo bbox from segmentation. If we get rid of object detection branch and perform segmentation on the entire image, the performance will be downgraded. We prove that with the ablation experiments MONet(ResNet18+FPN) and MONet(ResNet18+FPN+UNet).
> 5. Due to the lack of supervision signal, unsupervised object detection/instance segmentation is far from the supervised one. Most related works (MONet, IODINE(ICML 2019), GENESIS(ICLR2020), SPACE(ICLR2020)) only use supervision from scene reconstruction and their only difference is the network structure. Our novel self-supervision provides another signal which can be used to guide the other unsupervised scene decomposition.
> 6. To summary it, our work is entirely different from MONet in terms of inference and training. Our work also makes a non-trivial contribution to this topic by proposing a novel self-supervised method that can guide other unsupervised scene decomposition. We believe our model can not be treated as a slight variation of MONet. Otherwise, if following the same logic, GENESIS(ICLR2020) is just adding the recurrent connections between different latents into the MONet; SPACE(ICLR2020) is just combining SPAIR for the foreground and MONet for the background.

---

### Official Review · AnonReviewer1 · 2020-10-27
**A slight variation of existing models with modest performance boost and lacking analysis.**

**Rating:** 6
**Confidence:** 3

**Review:**

This paper presents a variation of the MONet model where an additional Region Proposal Network generates bounding boxes for various objects in the scene. An additional loss is introduced during training to make the segmentations produced by the MONet segmenter consistent with the proposed bounding boxes. Results are demonstrated on multi d-Sprites and CLEVR with modest performance gains.

The paper is somewhat middle of the road in most aspects - the proposed method is, in my opinion, only a slight variation on the existing MONet model. Though presented clearly, I don't feel that adding that loss makes the model better in any fundamental way, even if performance numbers are slightly better in some circumstances. Furthermore, though there is some ablation analysis, I feel the level of analysis of the results is sub-par - when a relatively simple variation of a model like here is proposed I would want to see an effort to analyse the contribution beyond how it affects the numbers - do we learn anything new by introducing the variation? does it tell us some fallacy or failure of the original model and if so, does it fix it? These are lacking here.

A few more concrete points:

* In Table 1 - why is the ResNet18 + FPN missing from d-Sprites dataset? This ablation is probably the single most important experiment present in the paper - I would want to see it reported on both datasets.
* In general - I feel the performance on these datasets is quite saturated and I hope to see results on more challenging data in the future - the proposed method included
* There is very little discussion about the choice of hyper-parameters in the paper - how were they chosen? is the system sensitive to these choices?

Post rebuttal comments:

Thank you authors for the detailed response - I think some of of my concerns have been answered - the paper may be a valid contribution to the community and I am raising my score.

---

> ### Author Response · Authors · 2020-11-18
> **Response to AnonReviewer1 (1/2)**
>
> 1. "In Table 1 - why is the ResNet18 + FPN missing from d-Sprites dataset? This ablation is probably the single most important experiment present in the paper - I would want to see it reported on both datasets."
>
> Thanks for pointing out. We add MONet(ResNet18+FPN) and R-MONet(ROI Align) on Multi-dSprites dataset. We also add MONet(ResNet18+FPN+UNet) on both datasets to prove that R-MONet(UNet)'s performance will downgrade if getting rid of the object detection branch and self-supervised method.
>
> 2. "In general - I feel the performance on these datasets is quite saturated and I hope to see results on more challenging data in the future - the proposed method included"
>
> We think the performance on these datasets is not saturated especially on the metrics mAP used to evaluate object detection. While +1% mAP can be seen as a huge improvement in supervised area, our model achieves +10% $mAP_{50:95}$, +10%$mAP_{50}$ and +20% $mAP_{75}$, comparing with the best unsupervised object detection model we can find (SPACE).
>
> Regarding segmentation performance, we still have non-trivial quantitative improvements over the baseline MONet, IODINE, and SPACE. This quantitative difference can also be verified through newly added qualitative figures.
>
> We agree that the proposed model still requires significant change to apply it to the real image dataset (the more challenging dataset you were referring to). But none of the previous models as we know can achieve acceptable performance on these datasets. In Sec.5 of the related work IODINE[8](accepted in ICML2019): "IODINE groups ImageNet not into meaningful objects but mostly into regions of similar color". In SPACE[19](accepted in ICLR2020): "Interesting future directions are to ... and to improve the model for natural images". The experiment only on synthetic datasets is widely adopted by SOTA(State-of-the-Art) works. Even though, comparing with previous models like AIR, SPAIR, GENESIS, and SPACE, the CLEVR dataset we use contains complex lighting effects and relatively closer to real images. The Multi-dSprites dataset contains complex shapes that challenge many related works such as IODINE and SPACE.
>
> 3. "There is very little discussion about the choice of hyper-parameters in the paper - how were they chosen? is the system sensitive to these choices?"
>
> IODINE,  SPACE,  MONet,  R-MONet(Lite), and  R-MONet(UNet)  are only sensitive to the VAE decoder scale (standard deviation). The best hyperparameter settings are reported in Appendix.B. The rest hyperparameters are robust in a reasonable range. We add this in Appendix.B in the revision.
>
> 4. "do we learn anything new by introducing the variation? does it tell us some fallacy or failure of the original model and if so, does it fix it? These are lacking here."
>
> We add a more detailed analysis in section4.1 in the revised paper. To summary it, MONet’s loss does not explicitly restrict multiple objects with similar colors existing in the same mask. As we can see in Figure (2, 4, 8), MONet can not separate close objects with similar colors. More than that, since MONet performs segmentation on the entire image, it often suffers from small objects. When two objects are visually connected with each other, MONet may group them together as a single object even with different colors. This case is shown in Figure (5, 6). We can also find in Figure (9, 10), MONet may split a single object in multiple masks under certain lighting effects such as reflection or shadow. On the contrary, both R-MONet(Lite) and R-MONet(UNet) will not have this problem.

---

> ### Author Response · Authors · 2020-11-18
> **Response to AnonReviewer1 (2/2)**
>
> 5. "the proposed method is, in my opinion, only a slight variation on the existing MONet model. Though presented clearly, I don't feel that adding that loss makes the model better in any fundamental way, even if performance numbers are slightly better in some circumstances."
>
> **The contribution of R-MONet and difference from MONet**
> 1. We want to highlight that R-MONet is entirely different from MONet in terms of both the inference part and the training part. The only common structure is the generation part.
> 2. Regarding the inference part, MONet is designed to use a recurrent attention module to extract object' masks one by one. Also, MONet performs segmentation on the entire image each time. It will increase the complexity of segmentation and hurt the segmentation quality, especially for connected small objects. We discuss this problem in the revised result analysis section. On the contrary, R-MONet extracts object' masks in parallel since R-MONet does not assume the objects in the scene are dependent. Regarding segmentation, R-MONet only performs segmentation inside ROI by introducing object detection branch. Since the segmentation is only performed around ROI, the segmentation quality is better and be able to eliminate the problems of MONet. Besides that, R-MONet can perform both unsupervised object detection and unsupervised instance segmentation (in a way entirely different from SPACE) while MONet can only perform unsupervised instance segmentation. We do not agree R-MONet is a slight variation of MONet just like Mask-RCNN is not a slight variation of UNet.
> 3. Regarding the training part, we add another supervision from localization inconsistency between predicted bbox and instance masks. We find MONet often split a single object into different masks or put multiple objects into the same mask (discussed in the revised result analysis section). This is caused by the defect of MONet's loss. MONet's loss does not explicitly restrict multiple objects with similar colors existing in the same mask. On the contrary, R-MONet will not have this problem. With the help of the Multi-Otsu thresholding method and loss from pseudo bbox, proposed self-supervision will split the ROIs which contain multiple objects. It is worth mentioning that previous jointly learning detection and instance segmentation(Mask-RCNN) only share some spirits with us. It is essentially different tasks receiving their own ground truth. Differently, our approach makes detection and instance segmentation supervise each other.
> 4. People may argue that the inference structure(ResNet18, FPN, RPN) we used is not new. At first, we want to point out that the network structure directly from the supervised learning topic does not perform very well in our tasks. R-MONet(Lite) (with the same inference structure as S4Net) does not surpass all our baselines. The old network structure needs to be carefully adapted and evaluated through experiments (well-known ROI Align significantly hurts the result). The best performance can only be achieved with our improved segmentation head (R-MONet(UNet)). This contribution is not trivial. Besides that, We want to highlight that the inference structure does not work without our proposed self-supervised method since object detection branch can not be trained without pseudo bbox from segmentation. If we get rid of object detection branch and perform segmentation on the entire image, the performance will be downgraded. We prove that with the ablation experiments MONet(ResNet18+FPN) and MONet(ResNet18+FPN+UNet).
> 5. Due to the lack of supervision signal, unsupervised object detection/instance segmentation is far from the supervised one. Most related works (MONet, IODINE(ICML 2019), GENESIS(ICLR2020), SPACE(ICLR2020)) only use supervision from scene reconstruction and their only difference is the network structure. Our novel self-supervision provides another signal which can be used to guide the other unsupervised scene decomposition.
> 6. To summary it, our work is entirely different from MONet in terms of inference and training. Our work also makes a non-trivial contribution to this topic by proposing a novel self-supervised method that can guide other unsupervised scene decomposition. We believe our model can not be treated as a slight variation of MONet. Otherwise, if following the same logic, GENESIS(ICLR2020) is just adding the recurrent connections between different latents into the MONet; SPACE(ICLR2020) is just combining SPAIR for the foreground and MONet for the background.

---

### Official Review · AnonReviewer4 · 2020-10-28

**Rating:** 3
**Confidence:** 3

**Review:**

SUMMARY

The paper presents a method to decompose scenes into its constituent objects. This is done with a generative framework that generates both bounding boxes and segmentation masks for each object. It relies on several previously existing technologies. Its main contribution is enforcing consistency between bounding boxes and segmentation masks.

PROS

* Outperforms the baselines.

CONS

* The paper can be hard to read.
* Contributions seem minor.
* Good results, but on two toy datasets only.

COMMENTS

The writing should be improved, as the paper can be hard to follow. One one hand, this includes broken sentences ("Inspired by the observation that foreground segmentation masks and bounding boxes both contain object geometric information and should be consistent with each other."), grammatical errors ("It proves that there are still many useful information can be discovered in those unlabeled data."), and sentences which are just hard to parse ("In the former type of models, the scene is encoded into the object-oriented disentangled spatial and appearance encoding explicitly."). On the other, the authors cite many concepts without introducing them in the paper (stick breaking, spatial broadcast decoder, multi-otsu thresholding, etc) which makes it non self-contained.

The paper presents what seem like engineering improvements over previous works (e.g. combining bounding boxes and segmentation masks) by adding more components to the framework, which is quite convoluted (see Fig. 1: ResNet, FPN, RPN, segmentation, VAEs, etc). It is hard to know where performance comes from, despite the ablation tests.

The experiments are limited to two toy datasets with a fixed number of simple objects (which must be known beforehand), which show no background interference and little occlusion.

In all, I do not think it meets the ICLR bar.

I am not an expert on the topic so I may have missed relevant datasets/baselines.

Detail: "Region of Interest" introduced after ROI has been mentioned several times.

---

> ### Author Response · Authors · 2020-11-18
> **Response to AnonReviewer4 (1/2)**
>
> 1. "The writing should be improved, as the paper can be hard to follow. One one hand, this includes broken sentences ("Inspired by the observation that foreground segmentation masks and bounding boxes both contain object geometric information and should be consistent with each other."), grammatical errors ("It proves that there are still many useful information can be discovered in those unlabeled data."), and sentences which are just hard to parse ("In the former type of models, the scene is encoded into the object-oriented disentangled spatial and appearance encoding explicitly.")."
>
> Thank you for pointing out that. We will update them in the revised paper.
>
> 2. "the authors cite many concepts without introducing them in the paper (stick breaking, spatial broadcast decoder, multi-otsu thresholding, etc) which makes it non self-contained."
>
> The stick-breaking process is a well-known analogy in statistics. We used equations 4-7 in the appendix to explain that. We explained the Multi-Otsu Thresholding method in the appendix. We add more details about it in the revised paper. Spatial broadcast decoder is almost the default VAE decoder used in the unsupervised scene decomposition topic such as MONet, IODINE(ICML2019), GENESIS(ICLR2020), SPACE(ICLR2020). We add some explanation about it in sec.2 of the revised paper.
>
> 3. "The experiments are limited to two toy datasets with a fixed number of simple objects (which must be known beforehand), which show no background interference and little occlusion."
>
> The datasets we used is widely adopted by many related works such as MONet and IODINE. Comparing with previous models like AIR, SPAIR, GENESIS, and SPACE, the CLEVR dataset we use contains complex lighting effects and relatively closer to real images. However, we want to mention that the unsupervised scene decomposing task on the real image dataset is still a challenging problem and none of the previous models as we know can achieve acceptable performance on these datasets. In Sec.5 of the related work IODINE[8](accepted in ICML2019): "IODINE groups ImageNet not into meaningful objects but mostly into regions of similar color". In SPACE[19](accepted in ICLR2020): "Interesting future directions are to ... and to improve the model for natural images". Experiment only on synthetic datasets is widely adopted by SOTA(State-of-the-Art) works.

---

> ### Author Response · Authors · 2020-11-18
> **Response to AnonReviewer4 (2/2)**
>
> 4. "The paper presents what seem like engineering improvements over previous works (e.g. combining bounding boxes and segmentation masks) by adding more components to the framework, which is quite convoluted (see Fig. 1: ResNet, FPN, RPN, segmentation, VAEs, etc). It is hard to know where performance comes from, despite the ablation tests."
>
> **The contribution of R-MONet and difference from MONet**
> 1. We want to highlight that R-MONet is entirely different from MONet in terms of both the inference part and the training part. The only common structure is the generation part.
> 2. Regarding the inference part, MONet is designed to use a recurrent attention module to extract object' masks one by one. Also, MONet performs segmentation on the entire image each time. It will increase the complexity of segmentation and hurt the segmentation quality, especially for connected small objects. We discuss this problem in the revised result analysis section. On the contrary, R-MONet extracts object' masks in parallel since R-MONet does not assume the objects in the scene are dependent. Regarding segmentation, R-MONet only performs segmentation inside ROI by introducing object detection branch. Since the segmentation is only performed around ROI, the segmentation quality is better and be able to eliminate the problems of MONet. Besides that, R-MONet can perform both unsupervised object detection and unsupervised instance segmentation (in a way entirely different from SPACE) while MONet can only perform unsupervised instance segmentation. We do not agree R-MONet is a slight variation of MONet just like Mask-RCNN is not a slight variation of UNet.
> 3. Regarding the training part, we add another supervision from localization inconsistency between predicted bbox and instance masks. We find MONet often split a single object into different masks or put multiple objects into the same mask (discussed in the revised result analysis section). This is caused by the defect of MONet's loss. MONet's loss does not explicitly restrict multiple objects with similar colors existing in the same mask. On the contrary, R-MONet will not have this problem. With the help of the Multi-Otsu thresholding method and loss from pseudo bbox, proposed self-supervision will split the ROIs which contain multiple objects. It is worth mentioning that previous jointly learning detection and instance segmentation(Mask-RCNN) only share some spirits with us. It is essentially different tasks receiving their own ground truth. Differently, our approach makes detection and instance segmentation supervise each other.
> 4. People may argue that the inference structure(ResNet18, FPN, RPN) we used is not new. At first, we want to point out that the network structure directly from the supervised learning topic does not perform very well in our tasks. R-MONet(Lite) (with the same inference structure as S4Net) does not surpass all our baselines. The old network structure needs to be carefully adapted and evaluated through experiments (well-known ROI Align significantly hurts the result). The best performance can only be achieved with our improved segmentation head (R-MONet(UNet)). This contribution is not trivial. Besides that, We want to highlight that the inference structure does not work without our proposed self-supervised method since object detection branch can not be trained without pseudo bbox from segmentation. If we get rid of object detection branch and perform segmentation on the entire image, the performance will be downgraded. We prove that with the ablation experiments MONet(ResNet18+FPN) and MONet(ResNet18+FPN+UNet).
> 5. Due to the lack of supervision signal, unsupervised object detection/instance segmentation is far from the supervised one. Most related works (MONet, IODINE(ICML 2019), GENESIS(ICLR2020), SPACE(ICLR2020)) only use supervision from scene reconstruction and their only difference is the network structure. Our novel self-supervision provides another signal which can be used to guide the other unsupervised scene decomposition.
> 6. To summary it, our work is entirely different from MONet in terms of inference and training. Our work also makes a non-trivial contribution to this topic by proposing a novel self-supervised method that can guide other unsupervised scene decomposition. We believe our model can not be treated as a slight variation of MONet. Otherwise, if following the same logic, GENESIS(ICLR2020) is just adding the recurrent connections between different latents into the MONet; SPACE(ICLR2020) is just combining SPAIR for the foreground and MONet for the background.

---

### Author Response · Authors · 2020-11-19
**For All Reviewers**

We want to thank all the reviewers for providing insightful feedback. We have uploaded the revised paper. Our revision mainly focuses on the following parts:
1. We add MONet(ResNet18+FPN) and R-MONet(ROI Align) experiments on Multi-dSprites dataset to resolve some concerns of AnonReviewer1.
2. We add MONet(ResNet18+FPN+UNet) experiments on both datasets to prove that the performance gain of R-MONet(UNet) is not from the backbone. MONet(ResNet18+FPN+UNet) will get rid of object detection branch and self-supervised training in R-MONet(UNet). It also transforms the inference part from parallel(R-MONet) into recurrent(MONet).
3. We rewrite the result analysis and ablation study section in Sec.4.1. We also add more qualitative figures in the appendix to explain the defect of our baselines (MONet, SPACE and IODINE).
4. We added the qualitative result of R-MONet(UNet) **(without the self-supervised loss and object detection branch)** on CLEVR dataset in the latest revision Fig.20. This model performs segmentation on the entire image and generates object masks in parallel. The ARI is 0(less than 0.0001) since all object segmentations are in one mask. The spatial attention module is good at segmenting objects from the background but unable to separate objects from each other. This proves that the loss of MONet does not prevent multiple objects from showing up in the same mask and the effectiveness of proposed self-supervised loss.
5. We added the qualitative result of R-MONet(UNet) results on MS-COCO 2017 dataset in the latest revision Fig.21. We use the pretrained ResNet18+FPN on ImageNet provided in torchvision. Unfortunately, the proposed model can not achieve the same performance as the supervised models. The VAE used can not generate complex objects in the scene. It focuses more on the region which has high contrast with its surrounding areas.

We will respond to each reviewer's comments in detail. We believe we have addressed all concerns and looking forward to some feedback about our revised paper.

---

### Decision · Program_Chairs · 2021-01-07
**Final Decision**

**Decision:**

Reject

**Comment:**

The paper has good contributions to a challenging problem, leveraging a Faster-RCNN framework with a novel self-supervised learning loss. However reviewer 4 and other chairs (in calibration) considered that the paper does not meet the bar for acceptance. The other reviewers did not champion the paper either, hence i am proposing rejection.

Pros:
- R1 and R3 agree that the proposed model improves over related models such as MONET.
- The value of the proposed self-supervised loss connecting bounding boxes and segmentations is well validated in experiments.

Cons:
- R4 gives good suggestions that may be useful to reach a broader readership, namely introducing more of the concepts used in the paper., e.g. "stick breaking, spatial broadcast decoder, multi-otsu thresholding" so it becomes more self-contained. R4 also suggests improving the writing more generally.
- R4 still finds the proposed "method quite complex yet derivative" after the rebuttal.
- All reviewers complain about lack of experiments in real data, but the authors did revise their paper and add some coco results in the appendix. These could be part of the main paper in a future version.